# SCALING CONVEX NEURAL NETWORKS WITH BURER-MONTEIRO FACTORIZATION

## ABSTRACT

Recently, it has been demonstrated that the training problem for a wide variety of (non) linear two-layer neural networks (such as two-layer perceptrons, convolutional networks, and self-attention) can be posed as equivalent convex optimization problems, with an induced regularizer which encourages low rank. However, this regularizer becomes prohibitively expensive to compute at moderate scales, impeding training convex neural networks. To this end, we propose applying the Burer-Monteiro factorization to convex neural networks, which for the first time enables a Burer-Monteiro perspective on neural networks with non-linearities. This factorization leads to an equivalent yet computationally tractable non-convex alternative with no spurious local minima. We develop a novel relative optimality bound of stationary points of the Burer-Monteiro factorization, thereby providing verifiable conditions under which any stationary point is a global optimum. Further, for the first time, we show that linear self-attention with sufficiently many heads has no spurious local minima. Our experiments demonstrate the implications of the relative optimality bound for stationary points of the Burer-Monteiro factorization.

## 1 INTRODUCTION

It has been demonstrated that the training problem for (non-linear) two-layer neural networks are equivalent to convex programs (Pilanci & Ergen, 2020; Ergen & Pilanci, 2020; Sahiner et al., 2021b; Ergen et al., 2021; Sahiner et al., 2021a). This has been observed for a variety of architectures, including multi-layer perceptrons (MLPs) (Pilanci & Ergen, 2020; Sahiner et al., 2021b), convolutional neural networks (CNNs) (Ergen & Pilanci, 2020; Sahiner et al., 2021c), and self-attention based transformers (Sahiner et al., 2022). A major benefit of convex training of neural networks is that global optimality is guaranteed, which brings transparency to training neural networks.

The convex formulation of neural networks induces biases by regularization of the network weights. For linear activation, the convex model directly imposes nuclear-norm regularization which is well-known to encourage low-rank solutions (Recht et al., 2010). For ReLU activation, however, the convex model induces a type of nuclear norm which promotes sparse factorization while the left factor is *constrained* to an affine space (Sahiner et al., 2021b). This constrained nuclear-norm is NP-hard to compute. This impedes the utility of convex neural networks for ReLU activation.

To address this computational challenge, we seek a method which $(i)$ inherits the per-iteration complexity of non-convex training of neural network, and $(ii)$ inherits the optimality guarantees and transparency of convex training. To find a solution, we leverage the well-studied Burer-Monterio (BM) factorization (Burer & Monteiro, 2003), which was originally proposed as a heuristic method to improve the complexity of convex semi-definite programs (SDPs).

BM has been applied as an efficient solution strategy for problems ranging from matrix factorization (Zheng & Lafferty, 2016; Park et al., 2017; Ge et al., 2017; Gillis, 2017) to rank minimization (Mardani et al., 2013; Recht et al., 2010; Wang et al., 2017) and matrix completion (Mardani et al., 2015; Ge et al., 2017). BM has also been used for over-simplified neural networks such as (Kawaguchi, 2016; Haeffele & Vidal, 2017; Du & Lee, 2018), where optimality conditions for local minima are provided. However, no work has deployed BM factorization for practical non-linear neural networks, and no guarantees are available about the optimality of stationary points. This is likely because BM theory is not applicable to the standard non-convex ReLU networks due to non-linearity between layer weights.

Thus, our focus in this work is to adapt BM for practical two-layer (non-linear) convex neural networks. We consider three common architectures, namely MLPs, CNNs, and self-attention networks. For these scenarios, we develop verifiable relative optimality bounds for all local minima and stationary points, which are easy and interpretable. In light of these conditions, we identify useful insights about the nature of neural networks contributing to optimality. In particular, we observe that for self-attention networks all local minima coincide with the global optima if there are sufficiently many heads. The optimality guarantees also provide useful algorithmic insights, allowing one to verify whether the light-weight first-order methods such as SGD achieve the global optimum for the non-convex training of neural networks. Our experiments with image classification task indicate that this BM factorization enables layerwise training of convex CNNs, which allows for convex networks for the first time to match the performance of multi-layer end-to-end trained non-convex CNNs.

## 1.1 CONTRIBUTIONS

All in all, our contributions are summarized as follows:

- We propose the BM factorization for efficiently solving convex neural networks with ReLU activation for moderate and large scales. This is the first time BM theory has been applied to the non-linear neural network setting.

- We derive a novel bound on the relative optimality of the stationary points of the BM factorization for neural networks.

- Accordingly, we identify simple and verifiable conditions which guarantee a stationary point of the non-convex BM formulation achieves the global optimum of the convex neural network.

- We yield basic insights into the fundamental nature of neural networks that contribute to optimality; e.g. that linear self-attention has no spurious local minima if it has sufficiently many heads.

- Our experiments verify the proposed relative optimality bound of stationary points from the BM factorization, and uncovers cases where SGD finds saddle points, even in two-layer neural networks.

## 1.2 RELATED WORK

**Burer-Monteiro factorization.** The Burer-Monteiro (BM) factorization was first introduced in (Burer & Monteiro, 2003; 2005). There has been a long line of work studying the use of this factorization for solving SDPs (Boumal et al., 2016; Cifuentes & Moitra, 2019; Waldspurger & Waters, 2020; Erdogdu et al., 2021). In the rectangular matrix case, gradient descent converges to a global optimum of the matrix factorization problem with high probability for certain classes of matrices (Zheng & Lafferty, 2016). The BM factorization has been also studied in the rectangular case in more generic settings (Bach et al., 2008; Haeffele et al., 2014; Haeffele & Vidal, 2017).

**Nuclear norm and rank minimization.** The ability of nuclear norm regularization to induce low rank has been studied extensively in compressed sensing (Candès & Recht, 2009; Recht et al., 2010; Candès & Tao, 2010). BM factorization has been applied to scale up nuclear-norm minimization (Mardani et al., 2015; 2013). It has also been deployed for low-rank matrix factorization (Cabral et al., 2013; Zhu et al., 2017; Park et al., 2017; Ge et al., 2017). The results show that all second-order critical points of the BM factorization are global optima if certain qualification conditions are met.

**SGD for non-convex neural networks.** It has been shown that for over-parameterized two-layer linear networks, all local minima are global minima (Kawaguchi, 2016). Accordingly, a line of work has attempted to show that gradient descent or its modifications provably find local minima and escape saddle points (Ge et al., 2015; Lee et al., 2016; Jin et al., 2017; Daneshmand et al., 2018). However, these works assume Lipschitz gradients and Hessians of the non-convex objective, which is not typically satisfied. Another line of work shows that gradient descent converges to global optima for sufficiently highly over-parameterized neural networks, with either the parameter count being a high-order polynomial of the sample count (Du et al., 2018; 2019; Arora et al., 2019), or the network architecture being simple (Du & Lee, 2018). In practice, it has been empirically observed that SGD

can converge to local maxima, or get stuck in saddle points (Du et al., 2017; Ziyin et al., 2021). For unregularized matrix factorization, it has also recently been shown that randomly initialized gradient descent on BM factorization provably converges to global minima (Ye & Du, 2021).

**Convex neural networks.** ReLU neural networks have equivalent convex programs for training, such as networks with scalar outputs (Pilanci & Ergen, 2020), vector-outputs (Sahiner et al., 2021b), convolutional networks (Ergen & Pilanci, 2020; Sahiner et al., 2021c), polynomial-activation networks (Bartan & Pilanci, 2021), batch-norm based networks (Ergen et al., 2021), Wasserstein GANs (Sahiner et al., 2021a), and self-attention networks (Sahiner et al., 2022). Despite efforts in developing efficient solvers, convex networks are only effectively trainable at small scales (Bai et al., 2022; Mishkin et al., 2022). Our novelty is to adapt BM factorization as a fast and scalable solution for training convex networks, with simple, verifiable conditions for global optimality.

## 2 PRELIMINARIES

We denote $(\cdot)_+ := \max\{0, \cdot\}$ as the ReLU non-linearity. We use superscripts, say $\mathbf{A}^{(i_i, i_2)}$, to denote blocks of matrices, and brackets, say $\mathbf{A}[i_1, i_2]$, to denote elements of matrices. We let $\mathbf{1}$ be the vector of ones of appropriate size, and $\mathcal{B}_H$ be the unit $H$-norm ball, $\{\mathbf{u} : \|\mathbf{u}\|_H \leq 1\}$. Unless otherwise stated, let $F$ be a convex, differentiable function. We use $n$ to denote the number of samples, and $c$ to denote the output dimension of each network. All proofs are presented in the Appendix.

### 2.1 TWO-LAYER NEURAL NETWORKS AS CONVEX PROGRAMS

A line of work has demonstrated that two-layer neural networks are equivalent to convex optimization problems. We consider a data matrix $\mathbf{X} \in \mathbb{R}^{n \times d}$ and consider two-layer $\sigma$-activation network with $c$ outputs, $m$ neurons, weight-decay parameter $\beta > 0$ :

$$p^*_{MLP} := \min_{\substack{\mathbf{W}_1 \in \mathbb{R}^{d \times m} \\ \mathbf{W}_2 \in \mathbb{R}^{c \times m}}} F(\sigma(\mathbf{X}\mathbf{W}_1)\mathbf{W}_2^\top) + \frac{\beta}{2} \sum_{j=1}^{m} \|\mathbf{w}_{1j}\|_2^2 + \|\mathbf{w}_{2j}\|_2^2. \tag{1}$$

When $\sigma$ is a linear activation and $m \geq m^*$ for some $m^* \leq \min\{d, c\}$, this problem is equivalent to ((Rennie & Srebro, 2005), Section 2.2)

$$p^*_{LMLP} = \min_{\mathbf{Z} \in \mathbb{R}^{d \times c}} F(\mathbf{X}\mathbf{Z}) + \beta\|\mathbf{Z}\|_*, \tag{2}$$

whereas for a ReLU activation and $m \geq m^*$ for some unknown, problem-dependent $m^* \leq nc$ ((Sahiner et al., 2021b), Thm. 3.1),

$$p^*_{RMLP} = \min_{\mathbf{Z}_j \in \mathbb{R}^{d \times c}} F(\sum_{j=1}^{P} \mathbf{D}_j \mathbf{X} \mathbf{Z}_j) + \beta \sum_{j=1}^{P} \|\mathbf{Z}_j\|_{*, K_j}, \tag{3}$$

$$\mathbf{K}_j := (2\mathbf{D}_j - \mathbf{I}_n)\mathbf{X}$$

where $\{\mathbf{D}_j\}_{j=1}^P = \{\text{diag}(\mathbb{1}\{\mathbf{X}\mathbf{u} \geq 0\}) : \mathbf{u} \in \mathbb{R}^d\}$ enumerates the possible activation patterns generated from $\mathbf{X}$, and the number of such patterns satisfies $P \leq 2r \left(\frac{e(n-1)}{r}\right)^r$, where $r := \text{rank}(\mathbf{X})$ (Stanley et al., 2004; Pilanci & Ergen, 2020). The expression (3) also involves a constrained nuclear norm expression, which is defined as

$$\|\mathbf{Z}\|_{*, K} := \min_{t \geq 0} t \text{ s.t. } \mathbf{Z} \in t\mathcal{C} \tag{4}$$

$$\mathcal{C} := \text{conv}\{\mathbf{Z} = \mathbf{u}\mathbf{v}^\top : \mathbf{K}\mathbf{u} \geq 0, \|\mathbf{u}\|_2 \leq 1, \|\mathbf{v}\|_2 \leq 1\}.$$

This norm is a quasi-nuclear norm, which differs from the standard nuclear norm in that the factorization upon which it relies imposes a constraint on its left factors. In convex ReLU neural networks, this norm enforces the existence of $\{\mathbf{u}_k, \mathbf{v}_k\}$ such that $\mathbf{Z} = \sum_k \mathbf{u}_k \mathbf{v}_k^\top$ and $\mathbf{D}_j \mathbf{X} \mathbf{Z} = \sum_k (\mathbf{X}\mathbf{u}_k)_+ \mathbf{v}_k^\top$, and penalizes $\sum_k \|\mathbf{u}_k \mathbf{v}_k^\top\|_*$. This norm is NP-hard to compute. A variant of these ReLU activations, called *gated ReLU* activations, achieves the piecewise linearity of ReLU activations without enforcing the constraints (Fiat et al., 2019). Specifically, the ReLU gates are fixed to some $\{\mathbf{h}_j\}_{j=1}^P$ to form

$$\sigma(\mathbf{X}\mathbf{w}_{1j}) := \text{diag}(\mathbb{1}\{\mathbf{X}\mathbf{h}_j \geq 0\})(\mathbf{X}\mathbf{w}_{1j}) = \mathbf{D}_j \mathbf{X} \mathbf{w}_{1j}. \tag{5}$$

With gated ReLU activation, the equivalent convex program is given by ((Mishkin et al., 2022), Thm. 2.2; (Sahiner et al., 2022), e.q. (8))

$$p_{GMLP}^* = \min_{\mathbf{Z}_j \in \mathbb{R}^{d \times c}} F(\sum_{j=1}^P \mathbf{D}_j \mathbf{X} \mathbf{Z}_j) + \beta \sum_{j=1}^P \|\mathbf{Z}_j\|_*, \tag{6}$$

which thereby converts the constrained nuclear norm penalty to a standard nuclear norm penalty, thereby improving the complexity of the ReLU network. In addition to the multi-layer perceptron (MLP) formulation, two-layer ReLU-activation convolutional neural networks (CNNs) with global average pooling have been demonstrated to be equivalent to convex programs as well (Sahiner et al., 2021b;c; Ergen & Pilanci, 2020). The non-convex formulation is given by

$$p_{RCNN}^* := \min_{\substack{\mathbf{w}_{1j} \in \mathbb{R}^h \\ \mathbf{w}_{2j} \in \mathbb{R}^c}} \sum_{i=1}^n F(\sum_{j=1}^m \mathbf{w}_{2j} \mathbf{1}^\top (\mathbf{X}_i \mathbf{w}_{1j})_+) + \frac{\beta}{2} \sum_{j=1}^m \|\mathbf{w}_{1j}\|_2^2 + \|\mathbf{w}_{2j}\|_2^2, \tag{7}$$

where samples $\mathbf{X}_i \in \mathbb{R}^{K \times h}$ are represented by patch matrices, which hold a convolutional patch of size $h$ in each of their $K$ rows. In particular, each row of $\mathbf{X}_i$ contains the data a convolutional kernel would perform an inner-product with, and $h$ is the product of kernel dimensions while $K$ is the number of patches each kernel passes over. It has been shown (Sahiner et al., 2021b) that as long as $m \geq m^*$ where $m^* \leq nc$, this is equivalent to a convex program ((Sahiner et al., 2021b), Cor. 5.1)

$$p_{RCNN}^* = \min_{\mathbf{Z}_j \in \mathbb{R}^{h \times c}} \sum_{i=1}^n F((\sum_{j=1}^P \mathbf{1}^\top \mathbf{D}_j^{(i)} \mathbf{X}_i \mathbf{Z}_j)^\top) + \beta \sum_{j=1}^P \|\mathbf{Z}_j\|_{*, \mathrm{K}_j} \tag{8}$$

$$\mathbf{K}_j := (2\mathbf{D}_j - \mathbf{I}_{nK})\mathbf{X}, \ \mathbf{X} := \begin{bmatrix} \mathbf{X}_1 \\ \cdots \\ \mathbf{X}_n \end{bmatrix}$$

where $\{\mathbf{D}_j\}_{j=1}^P = \{\mathrm{diag}\,(\mathbb{1}\{\mathbf{X}\mathbf{u} \geq 0\}) : \mathbf{u} \in \mathbb{R}^h\}$ and $\mathbf{D}_j^{(i)} \in \mathbb{R}^{K \times K}$. Noting that $P \leq 2r\left(\frac{e(n-1)}{r}\right)^r$ and $r \leq h$, the only exponential dependence of $P$ is on $h$, which is typically fixed. Lastly, we review existing convexity results for self-attention transformers (Sahiner et al., 2022). We have the following non-convex objective for a single block of multi-head self-attention with $m$ heads, where $\mathbf{X}_i \in \mathbb{R}^{s \times d}$ with $s$ tokens and $d$ features

$$p_{SA}^* := \min_{\substack{\mathbf{W}_{1j} \in \mathbb{R}^{d \times d} \\ \mathbf{W}_{2j} \in \mathbb{R}^{d \times c}}} \sum_{i=1}^n F\left(\sum_{j=1}^m \sigma\left(\mathbf{X}_i \mathbf{W}_{1j} \mathbf{X}_i^\top\right) \mathbf{X}_i \mathbf{W}_{2j}\right) + \frac{\beta}{2} \sum_{j=1}^m \|\mathbf{W}_{1j}\|_F^2 + \|\mathbf{W}_{2j}\|_F^2, \tag{9}$$

for which a variety of objectives $F$ can be posed, including classification (e.g. $F$ incorporates global average pooling followed by softmax-cross-entropy with labels) or denoising (e.g. $F$ is a squared loss against a label matrix). In the linear activation case, as long as $m \geq m^*$, where $m^* \leq \min\{d^2, dc\}$, this is equivalent to ((Sahiner et al., 2022), Thm. 3.1)

$$p_{LSA}^* = \min_{\mathbf{Z} \in \mathbb{R}^{d^2 \times dc}} \sum_{i=1}^n F\left(\sum_{k=1}^d \sum_{\ell=1}^d \mathbf{G}_i[k, \ell] \mathbf{X}_i \mathbf{Z}^{(k, \ell)}\right) + \beta \|\mathbf{Z}\|_*, \tag{10}$$

where $\mathbf{G}_i := \mathbf{X}_i^\top \mathbf{X}_i$, $\mathbf{G}_i[k, l] \in \mathbb{R}$, and $\{\mathbf{Z}^{(k, \ell)} \in \mathbb{R}^{d \times c}\}$ are block matrices which form $\mathbf{Z}$. A similar formulation can be posed for ReLU and Gated ReLU activations. In this work, we show that these network architectures are amenable to the BM factorization.

## 2.2 THE BURER-MONTEIRO FACTORIZATION

First proposed by Burer & Monteiro (2003), the Burer-Monteiro (BM) factorization proposes to solve SDPs over some square matrix $\mathbf{Q}$ in terms of rectangular factors $\mathbf{R}$ where $\mathbf{Q}$ is substituted by $\mathbf{R}\mathbf{R}^\top$. It was first demonstrated that solving over $\mathbf{R}$ does not introduce spurious local minima, given $\mathrm{rank}(\mathbf{R}) \geq \mathrm{rank}(\mathbf{Q}^*)$ for optimal solution to the original SDP $\mathbf{Q}^*$ (Burer & Monteiro, 2005). In general, we seek applications where we optimize over a non-square matrix $\mathbf{Z}$, i.e.

$$p_{CVX}^* := \min_{\mathbf{Z} \in \mathbb{R}^{d \times c}} F(\mathbf{Z}) \tag{11}$$

for a convex, differentiable function $F$. One may approach this by factoring $\mathbf{Z} = \mathbf{U}\mathbf{V}^\top$, where $\mathbf{U} \in \mathbb{R}^{d \times m}$, $\mathbf{V} \in \mathbb{R}^{c \times m}$ for some arbitrary choice $m$. Then, we have an equivalent non-convex problem over $\mathbf{R} := \begin{bmatrix} \mathbf{U} \\ \mathbf{V} \end{bmatrix}$, for $f(\mathbf{R}) = F(\mathbf{U}\mathbf{V}^\top)$:

$$p_{CVX}^* = \min_{\mathbf{R}} f(\mathbf{R}). \tag{12}$$

Noting that (11) is convex over $\mathbf{R}\mathbf{R}^\top = \begin{bmatrix} \mathbf{U}\mathbf{U}^\top & \mathbf{U}\mathbf{V}^\top \\ \mathbf{V}\mathbf{U}^\top & \mathbf{V}\mathbf{V}^\top \end{bmatrix}$, one may apply directly the result of Burer & Monteiro (2005) to conclude that as long as $m \geq \mathrm{rank}(\mathbf{Z}^*)$, all local minima of (12) are global minima of (11) (see Appendix A.1). A major issue with these results is that $\mathrm{rank}(\mathbf{Z}^*)$ is not known a priori. Naively, one may simply choose $m \geq \min\{d, c\}$ and be assured that $m \geq \mathrm{rank}(\mathbf{Z}^*)$, but this approach is not satisfactory if further under-parameterization is desired. To address this issue, work from Bach et al. (2008) and Haeffele et al. (2014) demonstrates that all rank-deficient local minimizers of (12) achieve the global minimum $p_{CVX}^*$ (under mild conditions, see Appendix A.2).

A long line of work has analyzed the conditions where known non-convex optimization algorithms will converge to second-order critical points (local minima) (Ge et al., 2015; Jin et al., 2017; Daneshmand et al., 2018). Under the assumption of a bounded $f$ and its Hessian, a second-order critical point can be found by noisy gradient descent (Ge et al., 2015), or other second-order algorithms (Sun et al., 2015). Even vanilla gradient descent with random initialization has been demonstrated to almost surely converge to a local minimum for $f$ with Lipschitz gradient (Lee et al., 2016). However, if the gradient of $f$ is not Lipschitz-continuous, there are no guarantees that gradient descent will find a second-order critical point of (12): one may encounter a stationary point which is a saddle. For example, in the linear regression setting, i.e.

$$f(\mathbf{R}) = \|\mathbf{X}\mathbf{U}\mathbf{V}^\top - \mathbf{Y}\|_F^2, \tag{13}$$

the gradient of $f$ is Lipschitz continuous with respect to $\mathbf{U}$ when $\mathbf{V}$ is fixed and vice-versa, but not Lipschitz continuous with respect to $\mathbf{R}$ (Mukkamala & Ochs, 2019). Thus, one may not directly apply the results of Ge et al. (2015); Sun et al. (2015); Lee et al. (2016) in this case. Instead, we seek to understand the conditions under which stationary points to (12) correspond to global optima of (11). One such condition is given in Mardani et al. (2013; 2015).

**Theorem 2.1** (From (Mardani et al., 2013)). *Stationary points $\hat{\mathbf{U}}, \hat{\mathbf{V}}$ of the optimization problem*

$$p^* := \min_{\mathbf{U}, \mathbf{V}} \frac{1}{2} \|\mathbf{U}\mathbf{V}^\top - \mathbf{Y}\|_F^2 + \frac{\beta}{2} \left( \|\mathbf{U}\|_F^2 + \|\mathbf{V}\|_F^2 \right) \tag{14}$$

*correspond to global optima $\mathbf{Z}^* = \hat{\mathbf{U}}\hat{\mathbf{V}}^\top$ of the equivalent convex optimization problem*

$$p^* = \min_{\mathbf{Z}} \frac{1}{2} \|\mathbf{Z} - \mathbf{Y}\|_F^2 + \beta \|\mathbf{Z}\|_* \tag{15}$$

*provided that $\|\mathbf{Y} - \hat{\mathbf{U}}\hat{\mathbf{V}}^\top\|_2 \leq \beta$.*

## 3 BURER-MONTEIRO FACTORIZATION FOR CONVEX NEURAL NETWORKS

### 3.1 MLPs

We first seek to compare the convex formulations of the MLP training problem (2), (3), and (6) to their BM factorizations. We describe how to find the BM factorization for any convex MLP.

**Lemma 3.1.** *For any matrix $\mathbf{M} \in \mathbb{R}^{n \times d_c}$, let $f(\mathbf{U}, \mathbf{V}) := F(\mathbf{M}\mathbf{U}\mathbf{V}^\top)$ be a differentiable function. For any $\beta > 0$ and arbitrary vector norms $\|\cdot\|_R$ and $\|\cdot\|_C$, we define the Burer-Monteiro factorization*

$$p^* := \min_{\substack{\mathbf{U} \in \mathbb{R}^{d_c \times m} \\ \mathbf{V} \in \mathbb{R}^{d_r \times m}}} f(\mathbf{U}, \mathbf{V}) + \frac{\beta}{2} \left( \sum_{j=1}^{m} \|\mathbf{u}_j\|_C^2 + \|\mathbf{v}_j\|_R^2 \right). \tag{16}$$

*For the matrix norm $\|\cdot\|_D$ defined as*

$$\|\mathbf{Z}\|_D := \max_{\mathbf{R}} \mathrm{trace}(\mathbf{R}^\top \mathbf{Z}) \text{ s.t. } \mathbf{u}^\top \mathbf{R}\mathbf{v} \leq 1 \, \forall \mathbf{u} \in \mathcal{B}_C, \, \forall \mathbf{v} \in \mathcal{B}_R, \tag{17}$$

*the problem (16) is equivalent to the convex optimization problem*

$$p^* = \min_{\mathbf{Z} \in \mathbb{R}^{d_c \times d_r}} F(\mathbf{MZ}) + \beta \|\mathbf{Z}\|_D. \tag{18}$$

*Remark* 3.2. In the case of a linear MLP, $\mathbf{M} = \mathbf{X}$, $d_c = d$, $d_r = c$, and $\| \cdot \|_D = \| \cdot \|_*$, so using the definition of $\| \cdot \|_D$, in the corresponding BM factorization, $R = 2$ and $C = 2$ (Bach et al., 2008). For a gated ReLU network, the regularizer is still the nuclear norm, and thus the same $R = C = 2$ regularization appears in the BM factorization. In the case of the ReLU MLP, the nuclear norm is replaced by $\| \cdot \|_D = \sum_{j=1}^{P} \| \cdot_j \|_{*, \mathrm{K}_j}$, which in the BM factorization amounts to having the constraint $\mathbf{K}_j \mathbf{U}_j \geq \mathbf{0}$. We accordingly express the BM factorization of convex MLPs below.

$$p^*_{LMLP} = \min_{\substack{\mathbf{U} \in \mathbb{R}^{d \times m} \\ \mathbf{V} \in \mathbb{R}^{c \times m}}} F(\mathbf{XUV}^\top) + \frac{\beta}{2} \left( \|\mathbf{U}\|_F^2 + \|\mathbf{V}\|_F^2 \right) \tag{19}$$

$$p^*_{GMLP} = \min_{\substack{\mathbf{U}_j \in \mathbb{R}^{d \times m} \\ \mathbf{V}_j \in \mathbb{R}^{c \times m}}} F(\sum_{j=1}^{P} \mathbf{D}_j \mathbf{XU}_j \mathbf{V}_j^\top) + \frac{\beta}{2} \sum_{j=1}^{P} \left( \|\mathbf{U}_j\|_F^2 + \|\mathbf{V}_j\|_F^2 \right) \tag{20}$$

$$p^*_{RMLP} = \min_{\substack{\mathbf{U}_j \in \mathbb{R}^{d \times m} : (2\mathbf{D}_j - \mathbf{I}_n)\mathbf{XU}_j \geq \mathbf{0} \\ \mathbf{V}_j \in \mathbb{R}^{c \times m}}} F(\sum_{j=1}^{P} \mathbf{D}_j \mathbf{XU}_j \mathbf{V}_j^\top) + \frac{\beta}{2} \sum_{j=1}^{P} \left( \|\mathbf{U}_j\|_F^2 + \|\mathbf{V}_j\|_F^2 \right) \tag{21}$$

*To the best of our knowledge, (21) presents the first application of BM factorization to a non-linear neural network*, which is enabled by the convex model (3).

In the linear case, the BM factorization (19) is identical to the original non-convex formulation of a linear MLP with $m$ neurons. Furthermore, in the case of gated ReLU, the BM factorization when $m = 1$ is equivalent to the original non-convex formulation. However, for ReLU activation two-layer networks, the BM factorization even when $m = 1$ corresponds to a different (i.e. constrained, rather than ReLU activation) model than the non-convex formulation. While the original convex program is NP-hard, the computation of the cost function of the BM factorization is very simple. Thus, the per-iteration complexity of the BM factorization is much lower than for the convex ReLU MLP.

The BM factorizations of these convex MLPs are non-convex, hence finding a global minimum appears intractable. However, the following theorem demonstrates that as long as a rank-deficient local minimum to the BM factorization is obtained, it corresponds to a global optimum.

**Theorem 3.3.** *If $m \geq \mathrm{rank}(\mathbf{Z}^*)$, where $\mathbf{Z}^*$ is a minimizer of (18), all local minima of the BM factorization (16) are global minima. Furthermore, if $F$ is twice-differentiable, any rank-deficient local minimum $\hat{\mathbf{R}} := \begin{bmatrix} \hat{\mathbf{U}} \\ \hat{\mathbf{V}} \end{bmatrix}$ of (16) corresponds to a global minimizer $\mathbf{Z}^* = \hat{\mathbf{U}}\hat{\mathbf{V}}^\top$ of (18).*

This result demonstrates that these two-layer convex MLPs have no spurious local minima under mild conditions. However, there remains an algorithmic challenge: it is not straightforward to obtain a guaranteed local minima when the gradients of $f$ are not Lipschitz continuous. The following result provides a general condition under which stationary points of the (16) are global optima of (18).

**Theorem 3.4.** *For any non-negative objective function $F$, for a stationary $(\hat{\mathbf{U}}, \hat{\mathbf{V}})$ of (16) with corresponding $\hat{\mathbf{Z}} = \hat{\mathbf{U}}\hat{\mathbf{V}}^\top$ with objective $\hat{p}$ for (18), the relative optimality gap $\frac{\hat{p} - p^*}{p^*}$ satisfies*

$$\frac{\hat{p} - p^*}{p^*} \leq \left( \frac{\|\nabla_{\mathbf{Z}} F(\mathbf{M}\hat{\mathbf{Z}})\|_D^*}{\beta} - 1 \right)_+ \tag{22}$$

*where $\| \cdot \|_D^*$ is the dual norm of $\| \cdot \|_D$.*

In particular, this bound can be calculated by taking the gradient of the unregularized objective function, evaluated at candidate solution $\hat{\mathbf{Z}}$ to the convex problem (18), which is formed by the stationary point of BM problem (16). Intuitively, if the $\| \cdot \|_D^*$ norm of this solution is less than $\beta$, then by the subgradient condition, $\hat{\mathbf{Z}}$ is an optimal solution of (18). In the case of a linear MLP with $\mathbf{X} = \mathbf{I}_d$, $F$ a squared-loss objective, and $\|\nabla_{\mathbf{Z}} F(\mathbf{M}\hat{\mathbf{Z}})\|_D^* \leq \beta$, our result exactly replicates the result

of Theorem 2.1 from Mardani et al. (2013). Furthermore, when this condition is not exactly satisfied, (22) provides a novel result in the form of an optimality gap bound. To our knowledge, this is the first result that generalizes the optimality conditions for stationary points from any BM factorization of a neural network. This provides an easily computable bound after solving (16) which quantifies how close a solution is to the global minimum. In the case of a ReLU MLP, the relative optimality gap is given by

$$
\frac{\hat{p} - p^*}{p^*} \leq \left( \max_{\substack{j \in [P] \\ \mathbf{u} \in \mathcal{B}_2 \\ \mathbf{K}_j \mathbf{u} \geq 0}} \frac{1}{\beta} \| \nabla_{\mathbf{z}_j} F(\sum_{j'=1}^{P} \mathbf{D}_{j'} \mathbf{X} \hat{\mathbf{Z}}_{j'}) \mathbf{u} \|_2 - 1 \right)_+ .
\tag{23}
$$

Computing this quantity amounts to solving a cone-constrained PCA problem (Deshpande et al., 2014), which can be done in polynomial-time when $d$ is constant. We should note that some stationary points are clearly present in any problem, such as $(\hat{\mathbf{U}}, \hat{\mathbf{V}}) = (\mathbf{0}, \mathbf{0})$, so we cannot conclude that all stationary points are global optima. However, in certain cases, the optimality gap of stationary points (22) is always zero as we show next.

**Theorem 3.5.** *A stationary point $(\hat{\mathbf{U}}, \hat{\mathbf{V}})$ of (16) is a global minimizer of (18) if $R = C = 2$ and*

$$
\mathrm{rank}(\hat{\mathbf{U}}) = \mathrm{rank}(\hat{\mathbf{V}}) = \min\{d_c, d_r\}.
\tag{24}
$$

Thus, for linear and gated ReLU MLPs, we can ensure that if the Burer-Monteiro factorization achieves a stationary point with full rank, it is corresponds with the global optimum of the convex program. We now can further extend these results to CNNs and self-attention architectures.

## 3.2 CNNs

Before proceeding to explore the BM factorization in the context of two-layer CNNs, we first provide a new result on an equivalent convex program for two-layer ReLU CNNs with arbitrary linear pooling operations, which extends the results of Sahiner et al. (2021b); Ergen & Pilanci (2020) on Global Average Pooling CNNs. Define $\mathbf{P}_a \in \mathbb{R}^{a \times K}$ to be a linear pooling matrix which pools the $K$ spatial dimensions to an arbitrary size $a$. Then, we express the non-convex two-layer CNN problem as

$$
p_{CNN}^* := \min_{\substack{\mathbf{w}_{1j} \in \mathbb{R}^h \\ \mathbf{W}_{2j} \in \mathbb{R}^{c \times a}}} \sum_{i=1}^n F\left( \sum_{j=1}^m \mathbf{W}_{2j} \mathbf{P}_a \sigma(\mathbf{X}_i \mathbf{w}_{1j}) \right) + \frac{\beta}{2} \sum_{j=1}^m \|\mathbf{w}_{1j}\|_2^2 + \|\mathbf{W}_{2j}\|_F^2.
\tag{25}
$$

**Theorem 3.6.** *For $\beta > 0$ and ReLU activation $\sigma(\cdot) = (\cdot)_+$, if $m \geq m^*$ where $m^* \leq nac$, then (25) is equivalent to a convex optimization problem, given by*

$$
p_{CNN}^* = \min_{\mathbf{Z}_k \in \mathbb{R}^{h \times ac}} \sum_{i=1}^n F\left( \sum_{k=1}^P \begin{bmatrix} \mathrm{trace}(\mathbf{P}_a \mathbf{D}_k^{(i)} \mathbf{X}_i \mathbf{Z}_k^{(1)}) \\ \vdots \\ \mathrm{trace}(\mathbf{P}_a \mathbf{D}_k^{(i)} \mathbf{X}_i \mathbf{Z}_k^{(c)}) \end{bmatrix} \right) + \beta \sum_{k=1}^P \|\mathbf{Z}_k\|_{*, \mathrm{K}_k},
\tag{26}
$$

$$
\mathbf{K}_k := (2\mathbf{D}_k - \mathbf{I}_{nK}) \begin{bmatrix} \mathbf{X}_1 \\ \cdots \\ \mathbf{X}_n \end{bmatrix}, \quad \mathbf{Z}_k^{(c')} \in \mathbb{R}^{h \times a} \; \forall c' \in [c].
$$

Thus, we provide a novel result which characterizes two-layer CNNs with arbitrary linear pooling operations as a convex program. Similar results can be shown for the linear and gated-ReLU activation cases[1]. With this established, we present our main results on the BM factorization for CNNs.

**Lemma 3.7.** *The BM factorization of the convex CNN problem with ReLU activation is given as follows.*

$$
p_{RCNN}^* = \min_{\substack{\{\{\mathbf{u}_{jk} \in \mathbb{R}^h\}_{j=1}^m\}_{k=1}^P \\ \{\{\mathbf{V}_{jk} \in \mathbb{R}^{c \times a}\}_{j=1}^m\}_{k=1}^P \\ (2\mathbf{D}_k^{(i)} - \mathbf{I})\mathbf{X}_i \mathbf{u}_{jk} \geq 0}} \sum_{i=1}^n F\left( \sum_{k=1}^P \sum_{j=1}^m \mathbf{V}_{jk} \mathbf{P}_a \mathbf{D}_k^{(i)} \mathbf{X}_i \mathbf{u}_{jk} \right) + \frac{\beta}{2} \sum_{k=1}^P \sum_{j=1}^m \left( \|\mathbf{u}_{jk}\|_F^2 + \|\mathbf{V}_{jk}\|_F^2 \right)
$$

$$
\tag{27}
$$

---

[1]We examine linear and gated ReLU activations for CNNs in the Appendix.

The BM factorization closely resembles the original non-convex formulation (25). Generally, (27) inherits the results of Theorems (3.3), (3.4), and (3.5); we present one such corollary here.

**Corollary 3.7.1.** *A stationary point* $((\hat{\mathbf{u}}_{jk}, \hat{\mathbf{V}}_{jk})_{j=1}^m)_{k=1}^P$ *of (27) corresponds to a global minimizer* $\hat{\mathbf{Z}}_k = \sum_{j=1}^m \hat{\mathbf{u}}_{jk}\text{vec}\left(\hat{\mathbf{V}}_{jk}\right)^\top$ *of (26) provided that*

$$\|\sum_{i=1}^n \nabla_{\mathbf{Z}_k} F\left(\sum_{k'=1}^P \begin{bmatrix} \text{trace}(\mathbf{P}_a \mathbf{D}_{k'}^{(i)} \mathbf{X}_i \mathbf{Z}_{k'}^{(1)}) \\ \vdots \\ \text{trace}(\mathbf{P}_a \mathbf{D}_{k'}^{(i)} \mathbf{X}_i \mathbf{Z}_{k'}^{(c)}) \end{bmatrix}\right)\mathbf{u}\|_2 \leq \beta, \forall k \in [P], \forall \mathbf{u} \in \mathcal{B}_2 : (2\mathbf{D}_k^{(i)} - \mathbf{I})\mathbf{X}_i \mathbf{u} \geq 0.$$

(28)

### 3.3 Multi-Head Self-Attention

We now for the first time extend BM factorization theory to self-attention networks.

**Lemma 3.8.** *The BM factorization of the convex self-attention problem with linear activation[2] is given as follows.*

$$p_{LSA}^* = \min_{\substack{\mathbf{U}_j \in \mathbb{R}^{d \times d} \\ \mathbf{V}_j \in \mathbb{R}^{d \times c}}} \sum_{i=1}^n F\left(\sum_{j=1}^m \mathbf{X}_i \mathbf{U}_j \mathbf{X}_i^\top \mathbf{X}_i \mathbf{V}_j\right) + \frac{\beta}{2}\sum_{j=1}^m \|\mathbf{U}_j\|_F^2 + \|\mathbf{V}_j\|_F^2$$

(29)

In addition to inheriting all of the results of Theorems 3.3, 3.4, and 3.5, noting the equivalence of the BM factorization with the original non-convex program (9), we are the first to show conditions under which there are no spurious local minima for self-attention networks.

**Corollary 3.8.1.** *The linear-activation self-attention network (29) has no spurious local minima as long as the number of heads satisfies $m \geq m^*$ where $m^* \leq \min\{d^2, dc\}$. Furthermore, for any twice-differentiable objective $F$, if for any local minimum $(\hat{\mathbf{U}}_j, \hat{\mathbf{V}}_j)_{j=1}^m$ of (29), the matrix*

$$\hat{\mathbf{R}} := \begin{bmatrix} \text{vec}(\hat{\mathbf{U}}_1) & \cdots & \text{vec}(\hat{\mathbf{U}}_m) \\ \text{vec}(\hat{\mathbf{V}}_1) & \cdots & \text{vec}(\hat{\mathbf{V}}_m) \end{bmatrix} \in \mathbb{R}^{d(d+c) \times m}$$

(30)

*is rank-deficient, then this local minimum is also a global minimum of (10).*

## 4 Experimental Results: The Relative Optimality Gap Bound

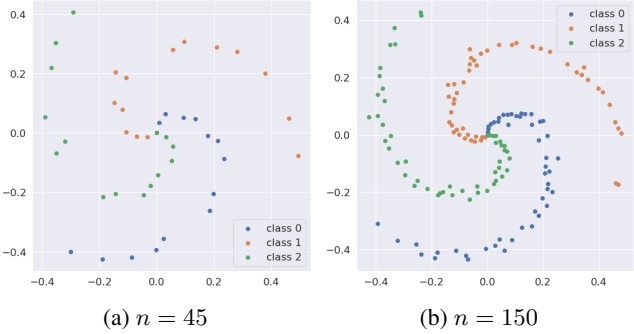

(a) $n = 45$          (b) $n = 150$

Figure 1: Example of three-class spiral dataset, with different number of samples $n$.

In this section, we illustrate the utility of our proposed relative optimality bound for stationary points in the setting of two-layer fully-connected networks. We also seek to examine how this bound changes with respect to the number of samples $n$, the regularization parameter $\beta$ (which controls the sparsity of the convex solution), and the number of factors in the BM factorization

---

[2]We examine gated ReLU and ReLU activations for self-attention in the Appendix.

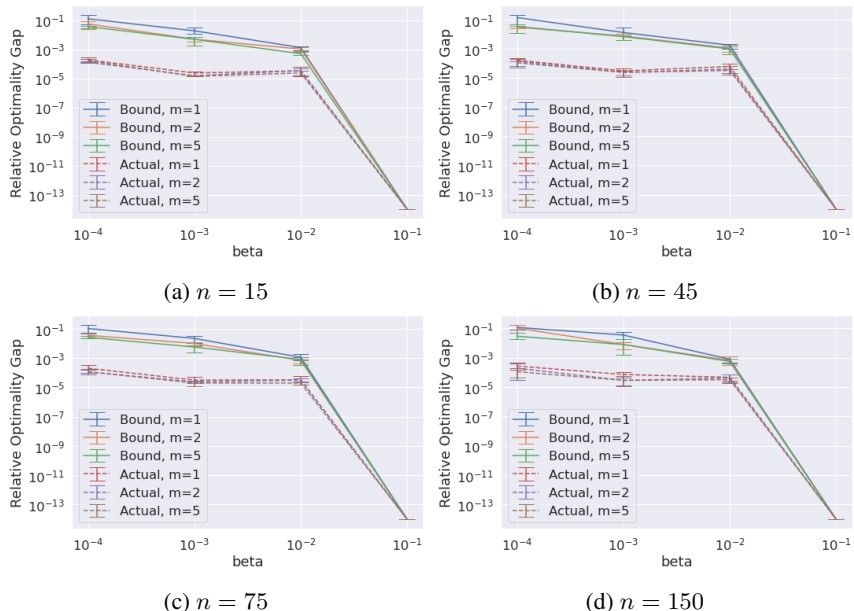

Figure 2: Relative optimality gap of the non-convex BM factorization of a gated-ReLU two-layer MLP for three-class spiral data classification ($d = 2$, $c = 3$). For fixed values of $n$, we demonstrate how $\beta$ and $m$ affect relative optimality gap, both in in terms of the proposed bound and the actual gap, where the global minimum is determined by convex optimization.

$m$. We initialize a class-balanced three-class spiral data set with varied number of samples $n$ (see Figure 1 for examples). For this dataset, we then train the gated ReLU MLP BM factorization (20) with varying number of factors $m$. We then compare the stationary points of these BM factorizations found by gradient descent (GD) to the global optimum, which we compute from (6).

For each stationary point of the BM factorization, we compute the relative optimality gap bound provided in our result in Theorem 3.4. We note that since $d = 2$, $c = 3$ in this case, for all $j$, $\text{rank}(\mathbf{Z}_j^*) \leq 2$, so as long as $m \geq 2$ all local minima of the BM factorization are global minima (Burer & Monteiro, 2005; Haeffele et al., 2014). While Lee et al. (2016) demonstrated that gradient descent with a random initialization converges to a local optimum almost surely for losses $f$ whose gradient is Lipschitz continuous, we use squared loss with one-hot-encoded class labels, for which $f$ is not Lipschitz continuous (Mukkamala & Ochs, 2019). Thus, there is no guarantee that GD will find the global minimum. We display results over $\beta$ for each fixed $n$ in Figure 2. For larger values of $\beta$, it becomes much easier for GD to find an optimal solution. We nevertheless find that our bound gives a useful proxy for whether the BM factorization has converged to the global minimum.

We find that GD applied to the BM factorization finds "subtle" saddle points: not quite local minima, but close. Interestingly, there is only a minor relationship between the optimality gap and the rank of the BM factorization $m$. While our optimality gap bound for $m = 1$ is larger than larger values of $m$ for small $\beta$, the actual optimality gap is nearly identical across $m$. This experiment further validates the need to consider stationary points of the BM factorization, rather than just local minima, to fully characterize the BM factorization for efficient solutions to convex problems.

## 5 CONCLUSION

We are the first to adapt the Burer-Monteiro (BM) factorization for two-layer convex neural networks with linear and ReLU activations, which offers new insights on their global optima. We provide a novel relative optimality bound on stationary point of the BM factorization, which provides a condition whose satisfaction guarantees a globally optimal solution.

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

# A  PROOFS

## A.1  RESULT OF (BURER & MONTEIRO, 2005) AND ITS APPLICATIONS TO RECTANGULAR MATRICES

In this subsection, we outline the precise theoretical statement of (Burer & Monteiro, 2005) and describe exactly how it corresponds to our summary in Section 2.2, and thus the application to the later derivations in our work. We first describe the following result, without proof, from (Burer & Monteiro, 2005).

**Lemma A.1** (Lemma 2.1 of (Burer & Monteiro, 2005)). *Suppose* $\mathbf{R} \in \mathbb{R}^{(d+c) \times r}$, $\mathbf{S} \in \mathbb{R}^{(d+c) \times r}$ *satisfy* $\mathbf{R}\mathbf{R}^\top = \mathbf{S}\mathbf{S}^\top$. *Then,* $\mathbf{S} = \mathbf{R}\mathbf{Q}$ *for some orthogonal* $\mathbf{Q} \in \mathbb{R}^{r \times r}$.

Now, we proceed to prove analogs of Theorem 2.3 of (Burer & Monteiro, 2005) for general SDPs with a rank constraint.

**Lemma A.2** (Analog of Lemma 2.2 of (Burer & Monteiro, 2005)). *Consider the problem*

$$\min_{\mathbf{R} \in \mathbb{R}^{(d+c) \times r}} F'(\mathbf{R}\mathbf{R}^\top). \tag{31}$$

$\hat{\mathbf{R}}$ *is a local minimum of (31) if and only if* $\hat{\mathbf{R}}\mathbf{Q}$ *is a local minimum of (31) for all orthogonal* $\mathbf{Q} \in \mathbb{R}^{r \times r}$.

*Proof.* Since $\mathbf{Q}$ is orthogonal, $(\hat{\mathbf{R}}\mathbf{Q})(\hat{\mathbf{R}}\mathbf{Q})^\top = \mathbf{R}\mathbf{Q}\mathbf{Q}^\top\mathbf{R}^\top = \mathbf{R}\mathbf{R}^\top$. Thus, $\hat{\mathbf{R}}' := \hat{\mathbf{R}}\mathbf{Q}$ attains the same objective value, gradients, and higher order derivatives as $\hat{\mathbf{R}}$. Thus, $\hat{\mathbf{R}}$ is a local minimum if and only if $\hat{\mathbf{R}}'$ is a local minimum. □

**Theorem A.3** (Analog of Theorem 2.3 of (Burer & Monteiro, 2005)). *Consider the following two problems.*

$$\min_{\substack{\mathbf{X} \in \mathbb{R}^{(d+c) \times (d+c)} \\ \mathbf{X} \succeq 0 \\ \text{rank}(\mathbf{X}) \leq r}} F'(\mathbf{X}), \tag{32}$$

$$\min_{\mathbf{R} \in \mathbb{R}^{(d+c) \times r}} F'(\mathbf{R}\mathbf{R}^\top). \tag{33}$$

*Then, for any continuous function* $F'$, *a feasible solution* $\hat{\mathbf{X}}$ *is a local minimizer of (32) if and only if, for* $\hat{\mathbf{X}} = \hat{\mathbf{R}}\hat{\mathbf{R}}^\top$, $\hat{\mathbf{R}}$ *is a local minimizer of (33).*

*Proof.* We follow the exact same lines as (Burer & Monteiro, 2005). By continuity of the map $\mathbf{R} \to \mathbf{R}\mathbf{R}^\top$, we know that if $\hat{\mathbf{X}}$ is a local minimizer of (32), then $\hat{\mathbf{R}}$ is a local minimizer of (33). Now, we must prove the other direction, namely that if $\hat{\mathbf{X}} = \hat{\mathbf{R}}\hat{\mathbf{R}}^\top$ is not local minimizer of (32), then $\hat{\mathbf{R}}$ is not a local minimizer of (33).

Suppose that $\hat{\mathbf{X}}$ is not a local minimum. By continuity of $F'$, then, there must be a sequence of feasible solutions $\{\mathbf{X}^k\}$ of (32) converging to $\hat{\mathbf{X}}$ such that $F'(\mathbf{X}^k) < F'(\hat{\mathbf{X}})$ for all $k$. For each $k$, choose $\mathbf{R}^k$ such that $\mathbf{X}^k = \mathbf{R}^k\mathbf{R}^{k^\top}$. Since $\{\mathbf{X}^k\}$ is bounded, it follows that $\{\mathbf{R}^k\}$ is bounded and hence has a subsequence $\{\mathbf{R}^k\}_{k \in \mathcal{K}}$ converging to some $\mathbf{R}$ such that $\hat{\mathbf{X}} = \mathbf{R}\mathbf{R}^\top$. Since $F'(\mathbf{R}^k\mathbf{R}^{k^\top}) = F'(\mathbf{X}^k) < F'(\hat{\mathbf{X}}) = F'(\mathbf{R}\mathbf{R}^\top)$, we see that $\mathbf{R}$ is not a local minimum of (33). Using the fact that $\hat{\mathbf{X}} = \hat{\mathbf{R}}\hat{\mathbf{R}}^\top = \mathbf{R}\mathbf{R}^\top$ together with Lemmas A.1 and A.2, we conclude that $\hat{\mathbf{R}}$ is not a local minimum of (33). □

With this established, we now describe how this theorem applies to the setting described in Section 2.2, i.e. the rectangular matrix, non-SDP case.

**Lemma A.4.** *Consider the solution* $\mathbf{X}^*$ *to*

$$p_1^* := \min_{\substack{\mathbf{X} \in \mathbb{R}^{(d+c) \times (d+c)} \\ \mathbf{X} \succeq 0}} F'(\mathbf{X}), \tag{34}$$

*and define $m^* := \text{rank}(\mathbf{X}^*)$. Then, for any $m \geq m^*$, (34) is equivalent to*

$$p_2^* := \min_{\substack{\mathbf{X} \in \mathbb{R}^{(d+c) \times (d+c)} \\ \mathbf{X} \succeq 0 \\ \text{rank}(\mathbf{X}) \leq m}} F'(\mathbf{X}), \tag{35}$$

*i.e. $p_1^* = p_2^*$, and the solutions to (35) and (34) are identical.*

*Proof.* Clearly, adding a rank constraint to (34) can only increase the objective value, so $p_2^* \geq p_1^*$. However, since the optimal solution to (34) satisfies the rank constraint for any $m \geq m^*$, every optimal solution of (34) maps to a feasible point for (35) that obtains the same objective, so $p_2^* \leq p_1^*$. Putting these together, we have $p_2^* = p_1^*$, and the solutions are identical. $\qquad\square$

**Lemma A.5.** *Consider the solution $\mathbf{X}^*$ to (34) with $m^* := \text{rank}(\mathbf{X}^*)$. Then, for any convex, continuous function $F'$, the global minimizer $\mathbf{X}^* = \hat{\mathbf{R}}\hat{\mathbf{R}}^\top$ corresponds to a local minimizer $\hat{\mathbf{R}}$ of*

$$\min_{\mathbf{R} \in \mathbb{R}^{(d+c) \times m}} F'(\mathbf{R}\mathbf{R}^\top), \tag{36}$$

*as long as $m \geq m^*$.*

*Proof.* We simply use the result from Lemma A.4 and apply the equivalence to Theorem A.3, noting that if $F'$ is convex, all local minimizers of (34) are global optimizers. $\qquad\square$

**Lemma A.6.** *Consider the optimization problem*

$$p_3^* := \min_{\mathbf{Z} \in \mathbb{R}^{d \times c}} F(\mathbf{Z}). \tag{37}$$

*Define $F' : \mathbb{R}^{(d+c) \times (d+c)} \to \mathbb{R}$ such that*

$$F'(\begin{bmatrix} \mathbf{X}_1 & \mathbf{X}_2 \\ \mathbf{X}_3 & \mathbf{X}_4 \end{bmatrix}) := F(\mathbf{X}_2) \tag{38}$$

*for $\mathbf{X}_1 \in \mathbb{R}^{d \times d}$, $\mathbf{X}_2 \in \mathbb{R}^{d \times c}$, $\mathbf{X}_3 \in \mathbb{R}^{c \times d}$, $\mathbf{X}_4 \in \mathbb{R}^{c \times c}$. Then, (37) is equivalent to (34), meaning that $p_1^* = p_3^*$ and their solutions map to each other.*

*Proof.* For any solution $\mathbf{X}^*$ to (34), we can simply form a solution $\mathbf{Z}^*$ to (37) by letting $\mathbf{Z}^* = \mathbf{X}_2^*$, so clearly $p_1^* \geq p_3^*$. For any solution $\mathbf{Z}^*$ to (37), factor $\mathbf{Z}^* = \mathbf{U}^*\mathbf{V}^{*\top}$ e.g. with SVD. Then, let $\mathbf{R}^* = \begin{bmatrix} \mathbf{U}^* \\ \mathbf{V}^* \end{bmatrix}$ and form a solution to (34) as $\mathbf{X}^* = \mathbf{R}^*\mathbf{R}^{*\top}$. Clearly, $\mathbf{X}^* \succeq 0$, so $\mathbf{X}^*$ is feasible, and the objective value is given by $F'(\mathbf{X}^*) = F(\mathbf{U}^*\mathbf{V}^{*\top}) = F(\mathbf{Z}^*) = p_1^*$. Thus, $p_3^* \leq p_1^*$. Putting these two together, we conclude $p_1^* = p_3^*$ and the solutions of the two problems map to each other. $\qquad\square$

**Lemma A.7** (Used in Section 2.2). *Consider the optimization problem (37), with optimal solution $\mathbf{Z}^*$ with $m^* := \text{rank}(\mathbf{Z}^*)$. Then, for any convex, continuous function $F'$, the solution $\mathbf{Z}^* = \hat{\mathbf{U}}\hat{\mathbf{V}}^\top$ corresponds to a local minimum $(\hat{\mathbf{U}}, \hat{\mathbf{V}})$ of*

$$\min_{\mathbf{U} \in \mathbb{R}^{d \times m}, \mathbf{V} \in \mathbb{R}^{c \times m}} F(\mathbf{U}\mathbf{V}^\top), \tag{39}$$

*as long as $m \geq m^*$.*

*Proof.* Define $F' : \mathbb{R}^{(d+c) \times (d+c)} \to \mathbb{R}$ such that

$$F'(\begin{bmatrix} \mathbf{X}_1 & \mathbf{X}_2 \\ \mathbf{X}_3 & \mathbf{X}_4 \end{bmatrix}) := F(\mathbf{X}_2) \tag{40}$$

for $\mathbf{X}_1 \in \mathbb{R}^{d \times d}$, $\mathbf{X}_2 \in \mathbb{R}^{d \times c}$, $\mathbf{X}_3 \in \mathbb{R}^{c \times d}$, $\mathbf{X}_4 \in \mathbb{R}^{c \times c}$. Then, let $\mathbf{R} = \begin{bmatrix} \mathbf{U} \\ \mathbf{V} \end{bmatrix} \in \mathbb{R}^{(d+c) \times m}$. One can re-write $F(\mathbf{U}\mathbf{V}^\top)$ as $F'(\mathbf{R}\mathbf{R}^\top)$. Then, we see that (39) can be expressed as (36), and any local minimizer to (36) is a local minimizer to (39).

Furthermore, noting from Lemma A.6, any optimal solution $\mathbf{Z}^* = \hat{\mathbf{U}}\hat{\mathbf{V}}^\top$ of (37) is equivalent to any optimal solution (34) by constructing $\mathbf{R}^* = \begin{bmatrix} \hat{\mathbf{U}} \\ \hat{\mathbf{V}} \end{bmatrix}$, $\mathbf{X}^* = \mathbf{R}^*\mathbf{R}^{*\top}$. Furthermore, $m^* = \text{rank}(\mathbf{Z}^*) = \text{rank}(\mathbf{R}^*) = \mathbf{X}^*$.

Putting everything together, we have that a rank-$m^*$ global optimum $\mathbf{Z}^* = \hat{\mathbf{U}}\hat{\mathbf{V}}^\top$ to (37) corresponds to a rank-$m^*$ global optimum $\mathbf{X}^*$ to (34) (Lemma A.6), which, as long as $m \geq m^*$, corresponds to a local minimum $\hat{\mathbf{R}} = \begin{bmatrix} \hat{\mathbf{U}} \\ \hat{\mathbf{V}} \end{bmatrix}$ (Lemma A.5), which is exactly the local minimum $(\hat{\mathbf{U}}, \hat{\mathbf{V}})$ of (39). $\qquad\square$

### A.2 RESULT OF (HAEFFELE ET AL., 2014) AND ITS APPLICATION TO RECTANGULAR MATRICES

In this subsection, we outline the precise theoretical statement of (Haeffele et al., 2014) and describe exactly how it corresponds to our summary in Section 2.2, and thus the application to the later derivations in our work. We first describe the following result, without proof, from (Haeffele et al., 2014).

**Theorem A.8** (Theorem 2 of (Haeffele et al., 2014)). *Let $F' : S_n^+ \to \mathbb{R}$ be of the form such that $F'(\mathbf{X}) = G'(\mathbf{X}) + H'(\mathbf{X})$, where $G'$ is convex, twice differentiable with compact level sets, and $H'$ is a proper convex function such that $F'$ is lower semi-continuous. Then, if $\hat{\mathbf{R}}$ is a rank-deficient local minimizer of*

$$\min_{\mathbf{R} \in \mathbb{R}^{(d+c) \times m}} F'(\mathbf{R}\mathbf{R}^\top), \tag{41}$$

*then $\mathbf{X}^* = \hat{\mathbf{R}}\hat{\mathbf{R}}^\top$ is a global minimizer of*

$$\min_{\substack{\mathbf{X} \in \mathbb{R}^{(d+c) \times (d+c)} \\ \mathbf{X} \succeq 0}} F'(\mathbf{X}). \tag{42}$$

We now describe how this theorem applies to the setting described in Section 2.2, i.e. the rectangular matrix, non-SDP case.

**Lemma A.9** (Used in Section 2.2). *Let $F : \mathbb{R}^{d \times c} \to \mathbb{R}$ be of the form such that $F(\mathbf{Z}) = G(\mathbf{Z}) + H(\mathbf{Z})$, where $G$ is convex, twice differentiable with compact level sets, and $H$ is a proper convex function such that $F$ is lower semi-continuous. Then, if $\hat{\mathbf{R}} = \begin{bmatrix} \hat{\mathbf{U}} \\ \hat{\mathbf{V}} \end{bmatrix}$ is a rank-deficient local minimizer of*

$$\min_{\substack{\mathbf{U} \in \mathbb{R}^{d \times m} \\ \mathbf{V} \in \mathbb{R}^{d \times m}}} F(\mathbf{U}\mathbf{V}^\top), \tag{43}$$

*then $\mathbf{Z}^* = \hat{\mathbf{U}}\hat{\mathbf{V}}^\top$ is a global minimizer of*

$$\min_{\mathbf{Z} \in \mathbb{R}^{d \times c}} F(\mathbf{Z}). \tag{44}$$

*Proof.* Define $F' : \mathbb{R}^{(d+c) \times (d+c)} \to \mathbb{R}$ such that

$$F'\left(\begin{bmatrix} \mathbf{X}_1 & \mathbf{X}_2 \\ \mathbf{X}_3 & \mathbf{X}_4 \end{bmatrix}\right) := F(\mathbf{X}_2) \tag{45}$$

for $\mathbf{X}_1 \in \mathbb{R}^{d \times d}$, $\mathbf{X}_2 \in \mathbb{R}^{d \times c}$, $\mathbf{X}_3 \in \mathbb{R}^{c \times d}$, $\mathbf{X}_4 \in \mathbb{R}^{c \times c}$. Clearly, if $F = G + H$, where $G$ is twice-differentiable and $H$ is proper convex, then, $F' = G' + H'$ where $G'$ is twice-differentiable and $H'$ is proper convex. From the proof of Lemma A.7, we know that (43) is exactly the same as (41) for $\mathbf{R} = \begin{bmatrix} \mathbf{U} \\ \mathbf{V} \end{bmatrix}$. Furthermore, we know from Lemma A.6 that any global minimum $\mathbf{Z}^*$ of (44) corresponds to a global minimum $\mathbf{X}^*$ of (42). Lastly, we know from Theorem A.8 that any rank-deficient local minimum of (41) corresponds to a global minimizer of (42).

Putting it all together, we have that a rank-deficient local minimizer $\hat{\mathbf{R}} = \begin{bmatrix} \hat{\mathbf{U}} \\ \hat{\mathbf{V}} \end{bmatrix}$ of (43) is a rank-deficient local minimizer of (41), which corresponds to a global optimizer $\mathbf{X}^* = \hat{\mathbf{R}}\hat{\mathbf{R}}^\top$ of (42) (Theorem A.8), which corresponds to a global optimizer $\mathbf{Z}^* = \hat{\mathbf{U}}\hat{\mathbf{V}}^\top$ of (44) (Lemma A.6). $\qquad\square$

### A.3 PROOF OF LEMMA 3.1

*Proof.* We first analyze the solution of the following optimization problem

$$f^* = \min_{\mathbf{u}_j, \mathbf{v}_j} \frac{1}{2} \left( \sum_{j=1}^m \|\mathbf{u}_j\|_C^2 + \|\mathbf{v}_j\|_R^2 \right) \text{ s.t. } \mathbf{U}\mathbf{V}^\top = \mathbf{Z}. \tag{46}$$

We can write this as an equivalent problem here (Bach et al., 2008; Pilanci & Ergen, 2020):

$$f^* = \min_{\mathbf{u}_j \in \mathcal{B}_C, \mathbf{v}_j} \left( \sum_{j=1}^m \|\mathbf{v}_j\|_R \right) \text{ s.t. } \mathbf{U}\mathbf{V}^\top = \mathbf{Z}. \tag{47}$$

We can form the Lagrangian of this as

$$f^* = \min_{\mathbf{u}_j \in \mathcal{B}_C, \mathbf{v}_j} \max_{\mathbf{R}} \left( \sum_{j=1}^m \|\mathbf{v}_j\|_R \right) - \text{trace}\left( \mathbf{R}^\top \mathbf{U}\mathbf{V}^\top \right) + \text{trace}\left( \mathbf{R}^\top \mathbf{Z} \right). \tag{48}$$

By Sion's minimax theorem, we can switch the maximum over $\mathbf{R}$ and minimum over $\mathbf{V}$ and minimize over $\mathbf{V}$ to obtain

$$f^* = \min_{\mathbf{u} \in \mathcal{B}_C} \max_{\mathbf{R}} \text{trace}\left( \mathbf{R}^\top \mathbf{Z} \right) \text{ s.t.} \|\mathbf{u}^\top \mathbf{R}\|_R^* \le 1. \tag{49}$$

As long as $m \ge \text{rank}(\mathbf{Z})$, by Slater's condition, we can switch the minimum and maximum (Shapiro, 2009) to obtain

$$f^* = \max_{\mathbf{R}} \text{trace}\left( \mathbf{R}^\top \mathbf{Z} \right) \text{ s.t.} \|\mathbf{u}^\top \mathbf{R}\|_R^* \le 1 \ \forall \mathbf{u} \in \mathcal{B}_C. \tag{50}$$

By the definition of dual norm, we can also write this as

$$f^* = \max_{\mathbf{R}} \text{trace}\left( \mathbf{R}^\top \mathbf{Z} \right) \text{ s.t.} \mathbf{u}^\top \mathbf{R}\mathbf{v} \le 1 \ \forall \mathbf{u} \in \mathcal{B}_C, \ \forall \mathbf{v} \in \mathcal{B}_R = \|\mathbf{Z}\|_D. \tag{51}$$

Thus, with this result, we have

$$p^* := \min_{\substack{\mathbf{U} \in \mathbb{R}^{d_c \times m} \\ \mathbf{V} \in \mathbb{R}^{d_r \times m}}} f(\mathbf{U}, \mathbf{V}) + \frac{\beta}{2} \left( \sum_{j=1}^m \|\mathbf{u}_j\|_C^2 + \|\mathbf{v}_j\|_R^2 \right), \tag{52}$$

equivalently as

$$p^* = \min_{\substack{\mathbf{U} \in \mathbb{R}^{d_c \times m} \\ \mathbf{V} \in \mathbb{R}^{d_r \times m}}} F(\mathbf{M}\mathbf{U}\mathbf{V}^\top) + \frac{\beta}{2} \left( \sum_{j=1}^m \|\mathbf{u}_j\|_C^2 + \|\mathbf{v}_j\|_R^2 \right), \tag{53}$$

or also as

$$p^* = \min_{\mathbf{Z}: \text{rank}(\mathbf{Z}) \le m} F(\mathbf{M}\mathbf{Z}) + \min_{\substack{\mathbf{U} \in \mathbb{R}^{d_c \times m} \\ \mathbf{V} \in \mathbb{R}^{d_r \times m} \\ \mathbf{U}\mathbf{V}^\top = \mathbf{Z}}} \frac{\beta}{2} \left( \sum_{j=1}^m \|\mathbf{u}_j\|_C^2 + \|\mathbf{v}_j\|_R^2 \right), \tag{54}$$

where we now apply our previous result to obtain

$$p^* = \min_{\mathbf{Z}: \text{rank}(\mathbf{Z}) \le m} F(\mathbf{M}\mathbf{Z}) + \beta \|\mathbf{Z}\|_D, \tag{55}$$

which if $\text{rank}(\mathbf{Z}^*) \ge m$ is equivalent to

$$p^* = \min_{\mathbf{Z}} F(\mathbf{M}\mathbf{Z}) + \beta \|\mathbf{Z}\|_D. \tag{56}$$

$$\square$$

## A.4 PROOF OF THEOREM 3.3

*Proof.* We simply note that (16) is the Burer-Monteiro factorization of (18). Thus, from Lemma A.7, as long as $m \geq \text{rank}(\mathbf{Z}^*)$, all local minima of (16) are global minima. Furthermore, note that (18) is composed of two components, one of which is a twice-differentiable function, and the other is a proper convex function. Thus, by Lemma A.9, all rank-deficient local minima are global minima. □

## A.5 PROOF OF THEOREM 3.4

*Proof.* Stationary points of (16) satisfy

$$\mathbf{0} \in \nabla_{\mathbf{U}} f(\hat{\mathbf{U}}, \hat{\mathbf{V}}) + \beta \|\hat{\mathbf{U}}\|_C \partial \|\hat{\mathbf{U}}\|_C$$
$$\mathbf{0} \in \nabla_{\mathbf{V}} f(\hat{\mathbf{U}}, \hat{\mathbf{V}}) + \beta \|\hat{\mathbf{V}}\|_R \partial \|\hat{\mathbf{V}}\|_R,$$

where we define

$$\|\hat{\mathbf{U}}\|_C := \sum_{j=1}^{m} \|\hat{\mathbf{u}}_j\|_C$$

and the same for $\|\hat{\mathbf{V}}\|_R$. By the definition of the subgradient, this stationarity condition can be written as

$$\exists \mathbf{U}' \text{ s.t. trace}(\hat{\mathbf{U}}^\top \mathbf{U}') = \|\hat{\mathbf{U}}\|_C, \|\mathbf{U}'\|_C^* \leq 1, \mathbf{0} = \nabla_{\mathbf{U}} f(\hat{\mathbf{U}}, \hat{\mathbf{V}}) + \beta \|\hat{\mathbf{U}}\|_C \mathbf{U}'$$
$$\exists \mathbf{V}' \text{ s.t. trace}(\hat{\mathbf{V}}^\top \mathbf{V}') = \|\hat{\mathbf{V}}\|_R, \|\mathbf{V}'\|_R^* \leq 1, \mathbf{0} = \nabla_{\mathbf{V}} f(\hat{\mathbf{U}}, \hat{\mathbf{V}}) + \beta \|\hat{\mathbf{V}}\|_R \mathbf{V}'.$$

By the chain rule, we have that

$$\mathbf{0} = \nabla_{\mathbf{Z}} F(\mathbf{M}\hat{\mathbf{Z}}) \hat{\mathbf{V}} + \beta \|\hat{\mathbf{U}}\|_C \mathbf{U}'$$
$$\mathbf{0} = \nabla_{\mathbf{Z}} F(\mathbf{M}\hat{\mathbf{Z}})^\top \hat{\mathbf{U}} + \beta \|\hat{\mathbf{V}}\|_R \mathbf{V}'$$

We now right-multiply the top equation by $\hat{\mathbf{U}}^\top$ and the bottom equation by $\hat{\mathbf{V}}^\top$ to obtain

$$\mathbf{0} = \nabla_{\mathbf{Z}} F(\mathbf{M}\hat{\mathbf{Z}}) \hat{\mathbf{V}} \hat{\mathbf{U}}^\top + \beta \|\hat{\mathbf{U}}\|_C \mathbf{U}' \hat{\mathbf{U}}^\top \tag{57}$$
$$\mathbf{0} = \nabla_{\mathbf{Z}} F(\mathbf{M}\hat{\mathbf{Z}})^\top \hat{\mathbf{U}} \hat{\mathbf{V}}^\top + \beta \|\hat{\mathbf{V}}\|_R \mathbf{V}' \hat{\mathbf{V}}^\top. \tag{58}$$

Taking the trace, we have

$$-\frac{1}{\beta} \text{trace} \left( \nabla_{\mathbf{Z}} F(\mathbf{M}\hat{\mathbf{Z}})^\top \hat{\mathbf{Z}} \right) = \|\hat{\mathbf{U}}\|_C \text{trace} \left( \mathbf{U}' \hat{\mathbf{U}}^\top \right) = \|\hat{\mathbf{U}}\|_R \text{trace} \left( \mathbf{V}' \hat{\mathbf{V}}^\top \right). \tag{59}$$

Noting the definitions of $\mathbf{U}'$ and $\mathbf{V}'$, we have

$$-\frac{1}{\beta} \text{trace} \left( \nabla_{\mathbf{Z}} F(\mathbf{M}\hat{\mathbf{Z}})^\top \hat{\mathbf{Z}} \right) = \|\hat{\mathbf{U}}\|_C^2 = \|\hat{\mathbf{V}}\|_R^2 \tag{60}$$

Furthermore, since clearly $F(\mathbf{M}\hat{\mathbf{Z}}) = f(\hat{\mathbf{U}}, \hat{\mathbf{V}})$, we have that

$$-\frac{1}{\beta} \text{trace} \left( \nabla_{\mathbf{Z}} F(\mathbf{M}\hat{\mathbf{Z}})^\top \hat{\mathbf{Z}} \right) = \|\hat{\mathbf{U}}\|_C^2 = \|\hat{\mathbf{V}}\|_R^2 = \frac{1}{2} (\|\hat{\mathbf{U}}\|_C^2 + \|\hat{\mathbf{V}}\|_R^2) = \|\hat{\mathbf{Z}}\|_D. \tag{61}$$

Now, we examine the optimality conditions for (18). In particular, we have

$$p^* = \min_{\mathbf{Z}} F(\mathbf{M}\mathbf{Z}) + \beta \|\mathbf{Z}\|_D, \tag{62}$$

which by definition of the dual norm, is equivalent to

$$p^* = \min_{\mathbf{Z}} \max_{\mathbf{Z}' \in \mathcal{B}_{D^*}} F(\mathbf{M}\mathbf{Z}) + \beta \text{trace}(\mathbf{Z}^\top \mathbf{Z}'). \tag{63}$$

Now suppose we have an approximate saddle point $(\hat{\mathbf{Z}}, \hat{\mathbf{Z}}')$ with objective value $\hat{p}$ such that $\nabla_{\mathbf{Z}} F(\hat{\mathbf{Z}}) + \beta \hat{\mathbf{Z}}' = 0$, $\text{trace}(\hat{\mathbf{Z}}^\top \hat{\mathbf{Z}}') = \|\hat{\mathbf{Z}}\|_D$, and $\hat{\mathbf{Z}}' \in (1 + \epsilon)\mathcal{B}_{D^*}$ for some $\epsilon \geq 0$. Let $\tilde{\mathbf{Z}} = \frac{1}{1+\epsilon} \hat{\mathbf{Z}}'$. By strong

duality and non-negativity of $F$, we have

$$p^* = \max_{\mathbf{Z}' \in \mathcal{B}_{D^*}} \min_{\mathbf{Z}} F(\mathbf{MZ}) + \beta \mathrm{trace}(\mathbf{Z}^\top \mathbf{Z}') \tag{64}$$

$$\geq \min_{\mathbf{Z}} F(\mathbf{MZ}) + \beta \mathrm{trace}(\mathbf{Z}^\top \tilde{\mathbf{Z}}) \tag{65}$$

$$= \min_{\mathbf{Z}} F(\mathbf{MZ}) + \frac{\beta}{1 + \epsilon} \mathrm{trace}(\mathbf{Z}^\top \hat{\mathbf{Z}}') \tag{66}$$

$$\geq \frac{1}{1 + \epsilon} \min_{\mathbf{Z}} \left( F(\mathbf{MZ}) + \beta \mathrm{trace}(\mathbf{Z}^\top \hat{\mathbf{Z}}') \right) \tag{67}$$

$$= \frac{1}{1 + \epsilon} \hat{p} \tag{68}$$

Rearranging, we have that

$$\frac{\hat{p} - p^*}{p^*} \leq \epsilon. \tag{69}$$

For the assumptions of this inequality to hold, for any candidate solution $\hat{\mathbf{Z}}$, one must satisfy $\nabla_{\mathbf{Z}} F(\mathbf{M}\hat{\mathbf{Z}}) + \beta \hat{\mathbf{Z}}' = 0$ and $\mathrm{trace}(\hat{\mathbf{Z}}^\top \hat{\mathbf{Z}}') = \|\hat{\mathbf{Z}}\|_D$. Solving the former equality for $\hat{\mathbf{Z}}' = -\frac{1}{\beta} \nabla_{\mathbf{Z}} F(\mathbf{M}\hat{\mathbf{Z}})$, we have by (61) that the latter equality is satisfied at any stationary point of the BM factorization. Lastly, for (69) to hold for a particular $\epsilon$, one must have

$$\hat{\mathbf{Z}}' = -\frac{1}{\beta} \nabla_{\mathbf{Z}} F(\mathbf{M}\hat{\mathbf{Z}}) \in (1 + \epsilon) \mathcal{B}_{D^*}, \tag{70}$$

so $\epsilon = \left( \frac{\|\nabla_{\mathbf{Z}} F(\mathbf{M}\hat{\mathbf{Z}})\|_D^*}{\beta} - 1 \right)_+$, i.e.

$$\frac{\hat{p} - p^*}{p^*} \leq \left( \frac{\|\nabla_{\mathbf{Z}} F(\mathbf{M}\hat{\mathbf{Z}})\|_D^*}{\beta} - 1 \right)_+ \tag{71}$$

$\square$

## A.6 Proof of Theorem 3.5

*Proof.* From the stationary point condition, we have

$$0 = \nabla_{\mathbf{Z}} F(\mathbf{M}\hat{\mathbf{Z}}) \hat{\mathbf{V}} + \beta \hat{\mathbf{U}} \tag{72}$$

$$0 = \nabla_{\mathbf{Z}} F(\mathbf{M}\hat{\mathbf{Z}})^\top \hat{\mathbf{U}} + \beta \hat{\mathbf{V}}. \tag{73}$$

From (73) we can obtain

$$\hat{\mathbf{U}}^\top \nabla_{\mathbf{Z}} F(\mathbf{M}\hat{\mathbf{Z}}) = -\beta \hat{\mathbf{V}}^\top. \tag{74}$$

Substituting this into (72), we have

$$0 = -\beta \hat{\mathbf{V}}^\top \hat{\mathbf{V}} + \beta \hat{\mathbf{U}}^\top \hat{\mathbf{U}} \tag{75}$$

$$\hat{\mathbf{V}}^\top \hat{\mathbf{V}} = \hat{\mathbf{U}}^\top \hat{\mathbf{U}}. \tag{76}$$

Thus, let $r := \mathrm{rank}(\hat{\mathbf{V}}) = \mathrm{rank}(\hat{\mathbf{U}}) \leq \min\{d_c, d_r\}$. We can write the compact SVD of the stationary point as

$$\hat{\mathbf{U}} = \mathbf{L}_{\mathbf{U}} \mathbf{\Lambda} \mathbf{R}^\top$$

$$\hat{\mathbf{V}} = \mathbf{L}_{\mathbf{V}} \mathbf{\Lambda} \mathbf{R}^\top,$$

where $\mathbf{L}_{\mathbf{U}} \in \mathbb{R}^{d_c \times r}$, $\mathbf{L}_{\mathbf{V}} \in \mathbb{R}^{d_r \times r}$. Assume without loss of generality that $d_c > d_r$, so $d_r = \min\{d_c, d_r\}$. We have

$$0 = \nabla_{\mathbf{Z}} F(\mathbf{M}\hat{\mathbf{Z}}) \hat{\mathbf{V}} + \beta \hat{\mathbf{U}} \tag{77}$$

$$0 = \nabla_{\mathbf{Z}} F(\mathbf{M}\hat{\mathbf{Z}}) \mathbf{L}_{\mathbf{V}} \mathbf{\Lambda} \mathbf{R}^\top + \beta \mathbf{L}_{\mathbf{U}} \mathbf{\Lambda} \mathbf{R}^\top \tag{78}$$

$$-\beta \mathbf{L}_{\mathbf{U}} = \nabla_{\mathbf{Z}} F(\mathbf{M}\hat{\mathbf{Z}}) \mathbf{L}_{\mathbf{V}}. \tag{79}$$

If $r = b$, $\mathbf{L_V}$ is square and therefore unitary, so

$$\|\nabla_{\mathbf{Z}}F(\mathbf{M}\hat{\mathbf{Z}})\|_2 = \|\nabla_{\mathbf{Z}}F(\mathbf{M}\hat{\mathbf{Z}})\mathbf{L_V}\|_2 \qquad (80)$$
$$= \| - \beta\mathbf{L_U}\|_2 \qquad (81)$$
$$= \beta. \qquad (82)$$

Thus, we satisfy (22) with equality. $\qquad\square$

In general, note that when $r < \min\{d_c, d_r\}$, we have

$$\|\nabla_{\mathbf{Z}}F(\mathbf{M}\hat{\mathbf{Z}})\|_2 = \|\nabla_{\mathbf{Z}}F(\mathbf{M}\hat{\mathbf{Z}})\|_2\|\mathbf{L_V}\|_2 \geq \|\nabla_{\mathbf{Z}}F(\mathbf{M}\hat{\mathbf{Z}})\mathbf{L_V}\|_2 \qquad (83)$$
$$= \| - \beta\mathbf{L_U}\|_2 \qquad (84)$$
$$= \beta, \qquad (85)$$

so (22) is a lower bound, which depends on how $\nabla_{\mathbf{Z}}F(\hat{\mathbf{Z}})$ behaves when operating on vectors in $\text{null}(\mathbf{L_V}^\top)$.

## A.7 PROOF OF THEOREM 3.6

*Proof.* We begin with the non-convex objective (25)

$$p^*_{CNN} := \min_{\substack{\mathbf{w}_{1j}\in\mathbb{R}^h \\ \mathbf{W}_{2j}\in\mathbb{R}^{c\times a}}} \sum_{i=1}^n F\left(\sum_{j=1}^m \mathbf{W}_{2j}\mathbf{P}_a(\mathbf{X}_i\mathbf{w}_{1j})_+\right) + \frac{\beta}{2}\sum_{j=1}^m \|\mathbf{w}_{1j}\|_2^2 + \|\mathbf{W}_{2j}\|_F^2. \qquad (86)$$

We can re-write this as (Bach et al., 2008; Pilanci & Ergen, 2020)

$$p^*_{CNN} = \min_{\substack{\mathbf{w}_{1j}\in\mathcal{B}_2 \\ \mathbf{W}_{2j}\in\mathbb{R}^{c\times a}}} \sum_{i=1}^n F\left(\sum_{j=1}^m \mathbf{W}_{2j}\mathbf{P}_a(\mathbf{X}_i\mathbf{w}_{1j})_+\right) + \beta\sum_{j=1}^m \|\mathbf{W}_{2j}\|_F. \qquad (87)$$

We can also re-write this as

$$p^*_{CNN} = \min_{\substack{\mathbf{w}_{1j}\in\mathcal{B}_2 \\ \mathbf{W}_{2j}\in\mathbb{R}^{c\times a} \\ \mathbf{r}_i}} \sum_{i=1}^n F\left(\mathbf{r}_i\right) + \beta\sum_{j=1}^m \|\mathbf{W}_{2j}\|_F \text{ s.t. } \sum_{j=1}^m \mathbf{W}_{2j}\mathbf{P}_a(\mathbf{X}_i\mathbf{w}_{1j})_+ = \mathbf{r}_i \qquad (88)$$

Forming the Lagrangian, we have

$$p^*_{CNN} = \min_{\substack{\mathbf{w}_{1j}\in\mathcal{B}_2 \\ \mathbf{W}_{2j}\in\mathbb{R}^{c\times a} \\ \mathbf{r}_i}} \max_{\mathbf{v}_i} \sum_{i=1}^n F\left(\mathbf{r}_i\right) + \beta\sum_{j=1}^m \|\mathbf{W}_{2j}\|_F + \sum_{i=1}^n \mathbf{v}_i^\top\left(\sum_{j=1}^m \mathbf{W}_{2j}\mathbf{P}_a(\mathbf{X}_i\mathbf{w}_{1j})_+ - \mathbf{r}_i\right) \qquad (89)$$

By Sion's minimax theorem, we can swap the minimum over $\mathbf{W}_{2j}, \mathbf{r}_i$ and maximum over $\mathbf{v}_i$, and minimize over $\mathbf{W}_{2j}, \mathbf{r}_i$ to obtain

$$p^*_{CNN} = \min_{\mathbf{u}\in\mathcal{B}_2} \max_{\mathbf{v}_i} -\sum_{i=1}^n -F^*(\mathbf{v}_i)$$

$$\text{s.t. } \|\sum_{i=1}^n \mathbf{P}_a(\mathbf{X}_i\mathbf{u})_+\mathbf{v}_i^\top\|_F \leq \beta \qquad (90)$$

where $F^*$ is the Fenchel conjugate of $F$. Now, as long as $\beta > 0$ and $m \geq m^*$ where $m^* \leq nac$, we can switch the order of max and min by Slater's condition (Shapiro, 2009; Sahiner et al., 2021b) to obtain

$$p^*_{CNN} = \max_{\mathbf{v}_i} -\sum_{i=1}^n -F^*(\mathbf{v}_i)$$

$$\text{s.t. } \max_{\mathbf{u}\in\mathcal{B}_2} \|\sum_{i=1}^n \mathbf{P}_a(\mathbf{X}_i\mathbf{u})_+\mathbf{v}_i^\top\|_F \leq \beta. \qquad (91)$$

Enumerating over the hyperplane arrangements $\{\mathbf{D}_k\}_{k=1}^P$, we can further write this as

$$p_{CNN}^* = \max_{\mathbf{v}_i} - \sum_{i=1}^n -F^*(\mathbf{v}_i)$$

$$\text{s.t.} \quad \max_{\substack{k \in [P] \\ \mathbf{u} \in \mathcal{B}_2 \\ (2\mathbf{D}_k^{(i)} - \mathbf{I})\mathbf{X}_i\mathbf{u} \geq 0}} \| \sum_{i=1}^n \mathbf{P}_a \mathbf{D}_k^{(i)} \mathbf{X}_i \mathbf{u}\mathbf{v}_i^\top \|_F \leq \beta. \tag{92}$$

Now, noting that $\text{vec}(\mathbf{ABC}) = (\mathbf{C}^\top \otimes \mathbf{A})\text{vec}(\mathbf{B})$ (Magnus & Neudecker, 2019), this is equivalent to

$$p_{CNN}^* = \max_{\mathbf{v}_i} - \sum_{i=1}^n F^*(\mathbf{v}_i)$$

$$\text{s.t.} \quad \max_{\substack{k \in [P] \\ \mathbf{u} \in \mathcal{B}_2 \\ (2\mathbf{D}_k^{(i)} - \mathbf{I})\mathbf{X}_i\mathbf{u} \geq 0}} \| \sum_{i=1}^n (\mathbf{v}_i \otimes \mathbf{P}_a \mathbf{D}_k^{(i)} \mathbf{X}_i)\mathbf{u}\|_2 \leq \beta \tag{93}$$

This may also be written further as

$$p_{CNN}^* = \max_{\mathbf{v}_i} - \sum_{i=1}^n F^*(\mathbf{v}_i)$$

$$\text{s.t.} \quad \max_{\substack{k \in [P] \\ \mathbf{u} \in \mathcal{B}_2 \\ (2\mathbf{D}_k^{(i)} - \mathbf{I})\mathbf{X}_i\mathbf{u} \geq 0 \\ \mathbf{g} \in \mathcal{B}_2}} \mathbf{g}^\top \sum_{i=1}^n (\mathbf{v}_i \otimes \mathbf{P}_a \mathbf{D}_k^{(i)} \mathbf{X}_i)\mathbf{u} \leq \beta, \tag{94}$$

and thereby as

$$p_{CNN}^* = \max_{\mathbf{v}_i} - \sum_{i=1}^n F^*(\mathbf{v}_i)$$

$$\text{s.t.} \quad \max_{\substack{k \in [P] \\ \mathbf{u} \in \mathcal{B}_2 \\ (2\mathbf{D}_k^{(i)} - \mathbf{I})\mathbf{X}_i\mathbf{u} \geq 0 \\ \mathbf{g} \in \mathcal{B}_2}} \text{trace}\left( \sum_{i=1}^n (\mathbf{v}_i \otimes \mathbf{P}_a \mathbf{D}_k^{(i)} \mathbf{X}_i)\mathbf{u}\mathbf{g}^\top \right) \leq \beta. \tag{95}$$

Now, we let $\mathbf{Z} = \mathbf{u}\mathbf{g}^\top$ to obtain

$$p_{CNN}^* = \max_{\mathbf{v}_i} - \sum_{i=1}^n F^*(\mathbf{v}_i)$$

$$\text{s.t.} \quad \max_{\substack{k \in [P] \\ \mathbf{Z} = \mathbf{u}\mathbf{g}^\top \\ \mathbf{u} \in \mathcal{B}_2 \\ (2\mathbf{D}_k^{(i)} - \mathbf{I})\mathbf{X}_i\mathbf{u} \geq 0 \\ \mathbf{g} \in \mathcal{B}_2}} \text{trace}\left( \sum_{i=1}^n (\mathbf{v}_i \otimes \mathbf{P}_a \mathbf{D}_k^{(i)} \mathbf{X}_i)\mathbf{Z} \right) \leq \beta. \tag{96}$$

We let $\mathcal{C}_k := \text{conv}\left\{ \mathbf{u}\mathbf{g}^\top : (2\mathbf{D}_k^{(i)} - \mathbf{I})\mathbf{X}_i\mathbf{u} \geq 0, \, \mathbf{u} \in \mathcal{B}_2, \, \mathbf{g} \in \mathcal{B}_2 \right\}$ and note that since our objective is linear we can take the convex hull of the constraints without changing the objective, to obtain

$$p_{CNN}^* = \max_{\mathbf{v}_i} - \sum_{i=1}^n F^*(\mathbf{v}_i)$$

$$\text{s.t.} \quad \max_{\substack{k \in [P] \\ \mathbf{Z} \in \mathcal{C}_k}} \text{trace}\left( \sum_{i=1}^n (\mathbf{v}_i \otimes \mathbf{P}_a \mathbf{D}_k^{(i)} \mathbf{X}_i)\mathbf{Z} \right) \leq \beta. \tag{97}$$

Note that the constraint $\mathbf{Z} \in \mathcal{C}_k$ is equivalent to stating that $\|\mathbf{Z}\|_{*,\mathrm{K}_k} \leq 1$ for the constrained nuclear norm definition with $\mathbf{K}_k = (2\mathbf{D}_k - \mathbf{I})\mathbf{X}$. Then, we have

$$p_{CNN}^* = \max_{\mathbf{v}_i} - \sum_{i=1}^{n} F^*(\mathbf{v}_i)$$

$$\text{s.t.} \max_{\substack{k \in [P] \\ \|\mathbf{Z}\|_{*,\mathrm{K}_k} \leq 1}} \text{trace}\left(\sum_{i=1}^{n} (\mathbf{v}_i \otimes \mathbf{P}_a \mathbf{D}_k^{(i)} \mathbf{X}_i)\mathbf{Z}\right) \leq \beta. \tag{98}$$

Now, we form the Lagrangian, given by

$$p_{CNN}^* = \max_{\mathbf{v}_i} \min_{\|\mathbf{Z}_k\|_{*,\mathrm{K}_k} \leq 1} \min_{\lambda_k \geq 0} -\sum_{i=1}^{n} F^*(\mathbf{v}_i) + \sum_{k=1}^{P} \lambda_k \left(\beta - \sum_{i=1}^{n} \text{vec}(\mathbf{Z}_k)^\top \mathbf{vec}\left(\mathbf{v}_i^\top \otimes (\mathbf{P}_a \mathbf{D}_k^{(i)} \mathbf{X}_i)^\top\right)\right). \tag{99}$$

By Sion's minimax theorem, we are permitted to change the order of the maxima and minima, to obtain

$$p_{CNN}^* = \min_{\lambda_k \geq 0} \min_{\|\mathbf{Z}_k\|_{*,\mathrm{K}_k} \leq 1} \max_{\mathbf{v}_i} -\sum_{i=1}^{n} F^*(\mathbf{v}_i) + \sum_{k=1}^{P} \lambda_k \left(\beta - \sum_{i=1}^{n} \text{vec}(\mathbf{Z}_k)^\top \mathbf{vec}\left(\mathbf{v}_i^\top \otimes (\mathbf{P}_a \mathbf{D}_k^{(i)} \mathbf{X}_i)^\top\right)\right). \tag{100}$$

Now, defining $\mathbf{K}_{a,1}$ as the $(a,1)$ commutation matrix we have the following identity from (Magnus & Neudecker, 2019):

$$\mathbf{vec}\left(\mathbf{v}_i^\top \otimes (\mathbf{P}_a \mathbf{D}_k^{(i)} \mathbf{X}_i)^\top\right) = \left(\mathbf{I}_c \otimes \left((\mathbf{K}_{a,1} \otimes \mathbf{I}_h)(\mathbf{vec}(\mathbf{X}_i^\top \mathbf{D}_k^{(i)} \mathbf{P}_a^\top))\right)\right) \mathbf{vec}(\mathbf{v}_i).$$

Using this identity and maximizing over $\mathbf{v}_i$, we obtain

$$p_{CNN}^* = \min_{\|\mathbf{Z}_k\|_{*,\mathrm{K}_k} \leq 1} \min_{\lambda_k \geq 0} \sum_{i=1}^{n} F\left(\sum_{k=1}^{P} \left(\mathbf{I}_c \otimes \left(\mathbf{vec}(\mathbf{X}_i^\top \mathbf{D}_k^{(i)} \mathbf{P}_a^\top)^\top (\mathbf{K}_{1,a} \otimes \mathbf{I}_h)\right)\right) \mathbf{vec}(\mathbf{Z}_k)\right) + \beta \sum_{k=1}^{P} \lambda_k. \tag{101}$$

Rescaling such that $\tilde{\mathbf{Z}}_k = \lambda_k \mathbf{Z}_k$, we obtain

$$p_{CNN}^* = \min_{\mathbf{Z}_k \in \mathbb{R}^{h \times ac}} \sum_{i=1}^{n} F\left(\sum_{k=1}^{P} \left(\mathbf{I}_c \otimes \left(\mathbf{vec}(\mathbf{X}_i^\top \mathbf{D}_k^{(i)} \mathbf{P}_a^\top)^\top (\mathbf{K}_{1,a} \otimes \mathbf{I}_h)\right)\right) \mathbf{vec}(\mathbf{Z}_k)\right) + \beta \sum_{k=1}^{P} \|\mathbf{Z}_k\|_{*,\mathrm{K}_k}. \tag{102}$$

Simplifying further, we can write this as

$$p_{CNN}^* = \min_{\mathbf{Z}_k \in \mathbb{R}^{h \times ac}} \sum_{i=1}^{n} F\left(\sum_{k=1}^{P} \begin{bmatrix} \text{trace}(\mathbf{P}_a \mathbf{D}_k^{(i)} \mathbf{X}_i \mathbf{Z}_k^{(1)}) \\ \vdots \\ \text{trace}(\mathbf{P}_a \mathbf{D}_k^{(i)} \mathbf{X}_i \mathbf{Z}_k^{(c)}) \end{bmatrix}\right) + \beta \sum_{k=1}^{P} \|\mathbf{Z}_k\|_{*,\mathrm{K}_k}, \tag{103}$$

where $\mathbf{Z}_k^{(c')} \in \mathbb{R}^{h \times a}$. $\qquad\square$

### A.8 PROOF OF LEMMA 3.7

*Proof.* We start with the convex formulation

$$p_{RCNN}^* = \min_{\mathbf{Z}_k \in \mathbb{R}^{h \times ac}} \sum_{i=1}^{n} F\left(\sum_{k=1}^{P} \begin{bmatrix} \text{trace}(\mathbf{P}_a \mathbf{D}_k^{(i)} \mathbf{X}_i \mathbf{Z}_k^{(1)}) \\ \vdots \\ \text{trace}(\mathbf{P}_a \mathbf{D}_k^{(i)} \mathbf{X}_i \mathbf{Z}_k^{(c)}) \end{bmatrix}\right) + \beta \sum_{k=1}^{P} \|\mathbf{Z}_k\|_{*,\mathrm{K}_k}. \tag{104}$$

In order to compute the Burer-Monteiro factorization, we factor $\mathbf{Z}_k = \mathbf{U}_k\mathbf{V}_k^\top$, where $\mathbf{U}_k \in \mathbb{R}^{h\times m}$, $\mathbf{V}_k \in \mathbb{R}^{ac\times m}$, and $(2\mathbf{D}_k^{(i)} - \mathbf{I})\mathbf{X}_i\mathbf{U}_k \geq \mathbf{0}$. Then, with $\mathbf{V}_k^{(c')} \in \mathbb{R}^{a\times m}$. Then for each $k$,

$$
\begin{bmatrix} \text{trace}(\mathbf{P}_a\mathbf{D}_k^{(i)}\mathbf{X}_i\mathbf{Z}_k^{(1)}) \\ \vdots \\ \text{trace}(\mathbf{P}_a\mathbf{D}_k^{(i)}\mathbf{X}_i\mathbf{Z}_k^{(c)}) \end{bmatrix} = \begin{bmatrix} \text{trace}(\mathbf{P}_a\mathbf{D}_k^{(i)}\mathbf{X}_i\mathbf{U}_k\mathbf{V}_k^{(1)^\top}) \\ \vdots \\ \text{trace}(\mathbf{P}_a\mathbf{D}_k^{(i)}\mathbf{X}_i\mathbf{U}_k\mathbf{V}_k^{(c)^\top}) \end{bmatrix}
$$
$$
= \begin{bmatrix} \text{trace}(\mathbf{V}_k^{(1)^\top}\mathbf{P}_a\mathbf{D}_k^{(i)}\mathbf{X}_i\mathbf{U}_k) \\ \vdots \\ \text{trace}(\mathbf{V}_k^{(c)^\top}\mathbf{P}_a\mathbf{D}_k^{(i)}\mathbf{X}_i\mathbf{U}_k) \end{bmatrix}
$$
$$
= \begin{bmatrix} \sum_{j=1}^m \mathbf{v}_{jk}^{(1)^\top}\mathbf{P}_a\mathbf{D}_k^{(i)}\mathbf{X}_i\mathbf{u}_{jk} \\ \vdots \\ \sum_{j=1}^m \mathbf{v}_{jk}^{(c)^\top}\mathbf{P}_a\mathbf{D}_k^{(i)}\mathbf{X}_i\mathbf{u}_{jk} \end{bmatrix}
$$
$$
= \sum_{j=1}^m \mathbf{V}_{jk}^\top\mathbf{P}_a\mathbf{D}_k^{(i)}\mathbf{X}_i\mathbf{u}_{jk},
$$

where

$$
\mathbf{V}_{jk}^\top := \begin{bmatrix} \mathbf{v}_{jk}^{(1)^\top} \\ \cdots \\ \mathbf{v}_{jk}^{(c)^\top} \end{bmatrix} \in \mathbb{R}^{c\times a}. \tag{105}
$$

The equivalent Burer-Monteiro formulation thus is given by

$$
p_{RCNN}^* = \min_{\substack{\{\{\mathbf{u}_{jk}\in\mathbb{R}^h\}_{j=1}^m\}_{k=1}^P \\ \{\{\mathbf{V}_{jk}\in\mathbb{R}^{c\times a}\}_{j=1}^m\}_{k=1}^P \\ (2\mathbf{D}_k^{(i)}-\mathbf{I})\mathbf{X}_i\mathbf{u}_{jk}\geq 0}} \sum_{i=1}^n F\left(\sum_{k=1}^P\sum_{j=1}^m \mathbf{V}_{jk}\mathbf{P}_a\mathbf{D}_k^{(i)}\mathbf{X}_i\mathbf{u}_{jk}\right) + \frac{\beta}{2}\sum_{k=1}^P\sum_{j=1}^m \left(\|\mathbf{u}_{jk}\|_F^2 + \|\mathbf{V}_{jk}\|_F^2\right). \tag{106}
$$

$\square$

## A.9 PROOF OF COROLLARY 3.7.1

*Proof.* We simply apply the result of Theorem 3.4, noting that stationary points correspond to global minima if the norm of the gradient is less than $\beta$. Thus, this condition is equivalent to

$$
\|\sum_{i=1}^n \nabla_{\mathbf{Z}_k} F\left(\sum_{k'=1}^P \begin{bmatrix} \text{trace}(\mathbf{P}_a\mathbf{D}_{k'}^{(i)}\mathbf{X}_i\mathbf{Z}_{k'}^{(1)}) \\ \vdots \\ \text{trace}(\mathbf{P}_a\mathbf{D}_{k'}^{(i)}\mathbf{X}_i\mathbf{Z}_{k'}^{(c)}) \end{bmatrix}\right)\mathbf{u}\|_2 \leq \beta, \ \forall k \in [P], \ \forall \mathbf{u} \in \mathcal{B}_2: \ (2\mathbf{D}_k^{(i)}-\mathbf{I})\mathbf{X}_i\mathbf{u} \geq 0. \tag{107}
$$

$\square$

## A.10 PROOF OF LEMMA 3.8

*Proof.* We begin from the convex formulation (10):

$$
p_{LSA}^* = \min_{\mathbf{Z}\in\mathbb{R}^{d^2\times dc}} \sum_{i=1}^n F\left(\sum_{k=1}^d\sum_{\ell=1}^d \mathbf{G}_i[k,\ell]\mathbf{X}_i\mathbf{Z}^{(k,\ell)}\right) + \beta\|\mathbf{Z}\|_*. \tag{108}
$$

Now, we seek to find the Burer-Monteiro factorization. We let $\mathbf{Z} = \mathbf{U}\mathbf{V}^\top$, where $\mathbf{U} \in \mathbb{R}^{d^2\times m}$ and $\mathbf{V} \in \mathbb{R}^{dc\times m}$. Let $\text{vec}^{-1}(\mathbf{u}_j) \in \mathbb{R}^{d\times d}$ be the result of taking chunks of $d$-length vectors from $\mathbf{u}_j$ for $j \in [m]$ and stacking them in columns. Similarly, let $\text{vec}^{-1}(\mathbf{v_j}) \in \mathbb{R}^{c\times d}$ be the result of

taking chunks of $c$-length vectors from $\mathbf{v}_j$ and stacking them in columns. Furthermore, we will let $\text{vec}^{-1}(\mathbf{u_j})_k$ be the $k$th column of $\text{vec}^{-1}(\mathbf{u_j})$. Then, recognize that

$$\mathbf{Z}^{(k,\ell)} = \sum_{j=1}^{m} \text{vec}^{-1}(\mathbf{u_j})_k \text{vec}^{-1}(\mathbf{v_j})_\ell^\top.$$

Thus,

$$\sum_{k=1}^{d} \sum_{\ell=1}^{d} \mathbf{G}_i[k,\ell] \mathbf{X}_i \mathbf{Z}^{(k,\ell)} = \sum_{k=1}^{d} \sum_{\ell=1}^{d} \sum_{j=1}^{m} \mathbf{G}_i[k,\ell] \mathbf{X}_i \text{vec}^{-1}(\mathbf{u_j})_k \text{vec}^{-1}(\mathbf{v_j})_\ell^\top$$

$$= \mathbf{X}_i \sum_{k=1}^{d} \sum_{\ell=1}^{d} \sum_{j=1}^{m} \text{vec}^{-1}(\mathbf{u_j})_k \mathbf{G}_i[k,\ell] \text{vec}^{-1}(\mathbf{v_j})_\ell^\top$$

$$= \mathbf{X}_i \sum_{j=1}^{m} \text{vec}^{-1}(\mathbf{u_j}) \mathbf{G}_i \text{vec}^{-1}(\mathbf{v_j})^\top$$

$$= \sum_{j=1}^{m} \mathbf{X}_i \text{vec}^{-1}(\mathbf{u_j}) \mathbf{X}_i^\top \mathbf{X}_i \text{vec}^{-1}(\mathbf{v_j})^\top.$$

Now, overloading notation, let $\text{vec}^{-1}(\mathbf{u_j}) = \mathbf{U}_j$ and $\text{vec}^{-1}(\mathbf{v_j})^\top = \mathbf{V}_j$. We have clearly that the Burer-Monteiro factorization of (10) is given by

$$p_{LSA}^* = \min_{\substack{\mathbf{U}_j \in \mathbb{R}^{d \times d} \\ \mathbf{V}_j \in \mathbb{R}^{d \times c}}} \sum_{i=1}^{n} F\left(\sum_{j=1}^{m} \mathbf{X}_i \mathbf{U}_j \mathbf{X}_i^\top \mathbf{X}_i \mathbf{V}_j\right) + \frac{\beta}{2} \sum_{j=1}^{m} \|\mathbf{U}_j\|_F^2 + \|\mathbf{V}_j\|_F^2. \tag{109}$$

$\square$

## A.11 Proof of Corollary 3.8.1

*Proof.* We simply nee dto apply the result of 3.3 to this setting. In this case, the non-convex linear self-attention network is equivalent to the Burer-Monteiro factorization of the convex form. To obtain this Burer-Monteiro factorization, we factorize convex weights $\mathbf{Z} \in \mathbb{R}^{d^2 \times dc}$, so $\text{rank}(\mathbf{Z}^*) \leq \min\{d^2, dc\}$. Thus, letting $m^* = \text{rank}(\mathbf{Z}^*) \leq \min\{d^2, dc\}$, we can observe that as long as the number of heads $m$ exceeds $m^*$, from Lemma A.7, all local optima are global. Further, we can form $\hat{\mathbf{R}} = \begin{bmatrix} \hat{\mathbf{U}} \\ \hat{\mathbf{V}} \end{bmatrix}$, and as long as this is a rank-deficient local minimum, it also corresponds to a global minimum when $F$ is twice-differentiable by Lemma A.9. $\square$

## B Additional Theoretical Results

### B.1 MLPs

The following theorem demonstrates that we can extend the results of (Sahiner et al., 2021b) beyond simply weight-decay regularization, and to arbitrary regularization.

**Theorem B.1.** *The non-convex ReLU training objective*

$$p^* := \min_{\mathbf{w}_{1j}, \mathbf{w}_{2j}} F(\sum_{j=1}^{m} (\mathbf{X}\mathbf{w}_{1j})_+ \mathbf{w}_{2j}) + \frac{\beta}{2} \left(\sum_{j=1}^{m} \|\mathbf{w}_{1j}\|_C^2 + \|\mathbf{w}_{2j}\|_R^2\right) \tag{110}$$

*is equivalent to the convex training objective*

$$p^* = \min_{\mathbf{Z}_j} F(\sum_{j=1}^{P} \mathbf{D}_j \mathbf{X} \mathbf{Z}_j) + \beta \sum_{j=1}^{P} \|\mathbf{Z}_j\|_{D_j}, \tag{111}$$

*as long as $\beta > 0$ and $m \geq m^*$ where $m^* \leq nc$, where*

$$\|\mathbf{Z}\|_{D_j} := \max_{\mathbf{R}} \text{trace}(\mathbf{R}^\top \mathbf{Z}) \text{ s.t. } \mathbf{u}^\top \mathbf{R} \mathbf{v} \leq 1 \, \forall \mathbf{u} \in \mathcal{B}_C : (2\mathbf{D}_j - \mathbf{I}_n)\mathbf{X}\mathbf{u} \geq 0, \, \forall \mathbf{v} \in \mathcal{B}_R. \tag{112}$$

*Proof.* We start by re-stating the convex objective (Bach et al., 2008; Pilanci & Ergen, 2020)

$$p^* := \min_{\substack{\mathbf{w}_{1j} \in \mathcal{B}_C \\ \mathbf{w}_{2j}}} F(\sum_{j=1}^{m} (\mathbf{X}\mathbf{w}_{1j})_+ \mathbf{w}_{2j}) + \beta \sum_{j=1}^{m} \|\mathbf{w}_{2j}\|_R. \tag{113}$$

Then, we can re-write this in a constrained form

$$p^* = \min_{\substack{\mathbf{w}_{1j} \in \mathcal{B}_C \\ \mathbf{w}_{2j} \\ \mathbf{R}}} F(\mathbf{R}) + \beta \sum_{j=1}^{m} \|\mathbf{w}_{2j}\|_R \text{ s.t.} \sum_{j=1}^{m} (\mathbf{X}\mathbf{w}_{1j})_+ \mathbf{w}_{2j} = \mathbf{R}, \tag{114}$$

and then the Lagrangian

$$p^* = \min_{\substack{\mathbf{w}_{1j} \in \mathcal{B}_C \\ \mathbf{w}_{2j} \\ \mathbf{R}}} \max_{\mathbf{V}} F(\mathbf{R}) + \beta \sum_{j=1}^{m} \|\mathbf{w}_{2j}\|_R + \text{trace} \left( \mathbf{V}^\top \left( \sum_{j=1}^{m} (\mathbf{X}\mathbf{w}_{1j})_+ \mathbf{w}_{2j} - \mathbf{R} \right) \right). \tag{115}$$

By Sion's minimax theorem, we can swap the order of the maximization over $\mathbf{V}$ and minimization over $\mathbf{w}_{2j}$ and $\mathbf{R}$. Then, minimizing over these two, we have

$$p^* = \min_{\mathbf{u} \in \mathcal{B}_C} \max_{\mathbf{V}} -F^*(\mathbf{V}) \text{ s.t.} \|\mathbf{V}^\top (\mathbf{X}\mathbf{u})_+\|_R^* \leq \beta. \tag{116}$$

By Slater's condition, which holds when $\beta > 0$ and $m \leq m^*$ where $m^* \leq nc$ (Shapiro, 2009; Sahiner et al., 2021c), we can switch the order of minimum and maximum to obtain

$$p^* = \max_{\mathbf{V}} -F^*(\mathbf{V}) \text{ s.t.} \max_{\mathbf{u} \in \mathcal{B}_C} \|\mathbf{V}^\top (\mathbf{X}\mathbf{u})_+\|_R^* \leq \beta. \tag{117}$$

Introducing hyperplane arrangements, we have

$$p^* = \max_{\mathbf{V}} -F^*(\mathbf{V}) \text{ s.t.} \max_{\substack{j \in [P] \\ \mathbf{u} \in \mathcal{B}_C \\ (2\mathbf{D}_j - \mathbf{I})\mathbf{X}\mathbf{u} \geq 0}} \|\mathbf{V}^\top \mathbf{D}_j \mathbf{X}\mathbf{u}\|_R^* \leq \beta. \tag{118}$$

By the concept of dual norm, this is equivalent to

$$p^* = \max_{\mathbf{V}} -F^*(\mathbf{V}) \text{ s.t.} \max_{\substack{j \in [P] \\ \mathbf{u} \in \mathcal{B}_C \\ (2\mathbf{D}_j - \mathbf{I})\mathbf{X}\mathbf{u} \geq 0 \\ \mathbf{g} \in \mathcal{B}_r}} \text{trace} \left( \mathbf{V}^\top \mathbf{D}_j \mathbf{X}\mathbf{u}\mathbf{g}^\top \right) \leq \beta. \tag{119}$$

Define

$$\|\mathbf{Z}\|_j := \max_{t \geq 0} t \text{ s.t.} \mathbf{Z} \in t\text{conv}\{\mathbf{u}\mathbf{g}^\top : \mathbf{u} \in \mathcal{B}_C, (2\mathbf{D}_j - \mathbf{I})\mathbf{X}\mathbf{u} \geq 0, \mathbf{g} \in \mathcal{B}_r\}. \tag{120}$$

Then, we can write our problem as

$$p^* = \max_{\mathbf{V}} -F^*(\mathbf{V}) \text{ s.t.} \max_{\substack{j \in [P] \\ \|\mathbf{Z}\|_j \leq 1}} \text{trace} \left( \mathbf{V}^\top \mathbf{D}_j \mathbf{X}\mathbf{Z} \right) \leq \beta. \tag{121}$$

Now, observe that $\|\mathbf{Z}\|_j = \|\mathbf{Z}\|_{D_j}$. In particular, let us examine (112), which we can re-write as

$$\|\mathbf{Z}\|_{D_j} = \max_{\mathbf{R}} \text{trace}(\mathbf{R}^\top \mathbf{Z}) \text{ s.t.} \max_{\|\mathbf{Z}\|_j \leq 1} \text{trace}(\mathbf{Z}^\top \mathbf{R}) \leq 1 \tag{122}$$

$$= \max_{\mathbf{R}} \text{trace}(\mathbf{R}^\top \mathbf{Z}) \text{ s.t.} \|\mathbf{R}\|_j^* \leq 1 \tag{123}$$

$$= \|\mathbf{Z}\|_j, \tag{124}$$

where the simplifications are made noting the definition of the dual norm. Now, we can write our objective as

$$p^* = \max_{\mathbf{V}} -F^*(\mathbf{V}) \text{ s.t.} \max_{\substack{j \in [P] \\ \|\mathbf{Z}\|_{D_j} \leq 1}} \text{trace} \left( \mathbf{V}^\top \mathbf{D}_j \mathbf{X}\mathbf{Z} \right) \leq \beta. \tag{125}$$

Forming the Largrangian, we have

$$p^* = \max_{\substack{\mathbf{V} \\ \lambda_j \geq 0}} \min_{\|\mathbf{Z}_j\|_{D_j} \leq 1} -F^*(\mathbf{V}) + \sum_{j=1}^{P} \lambda_j \left( \beta - \text{trace}\left( \mathbf{V}^\top \mathbf{D}_j \mathbf{X} \mathbf{Z}_j \right) \right). \tag{126}$$

By Sion's minimax theorem, we can switch max and min and solve over $\mathbf{V}$ to obtain

$$p^* = \min_{\substack{\|\mathbf{Z}_j\|_{D_j} \leq 1 \\ \lambda_j \geq 0}} F(\sum_{j=1}^{P} \lambda_j \mathbf{D}_j \mathbf{X} \mathbf{Z}_j) + \beta \sum_{j=1}^{P} \lambda_j. \tag{127}$$

Lastly, we can combine $\mathbf{Z}_j \lambda_j$ into one variable to obtain

$$p^* = \min_{\mathbf{Z}_j} F(\sum_{j=1}^{P} \mathbf{D}_j \mathbf{X} \mathbf{Z}_j) + \beta \sum_{j=1}^{P} \|\mathbf{Z}_j\|_{D_j}, \tag{128}$$

as desired. $\qquad\square$

## B.2 CNNs

**Lemma B.2.** *The Burer-Monteiro factorization of the convex CNN problem with linear and gated ReLU activation are given as follows.*

$$p_{LCNN}^* = \min_{\substack{\{\mathbf{u}_j \in \mathbb{R}^h\}_{j=1}^m \\ \{\mathbf{V}_j \in \mathbb{R}^{c \times a}\}_{j=1}^m}} \sum_{i=1}^{n} F\left( \sum_{j=1}^{m} \mathbf{V}_j \mathbf{P}_a \mathbf{X}_i \mathbf{u}_j \right) + \frac{\beta}{2} \sum_{j=1}^{m} \left( \|\mathbf{u}_j\|_2^2 + \|\mathbf{V}_j\|_F^2 \right) \tag{129}$$

$$p_{GCNN}^* = \min_{\substack{\{\{\mathbf{u}_{jk} \in \mathbb{R}^h\}_{j=1}^m\}_{k=1}^P \\ \{\{\mathbf{V}_{jk} \in \mathbb{R}^{c \times a}\}_{j=1}^m\}_{k=1}^P}} \sum_{i=1}^{n} F\left( \sum_{k=1}^{P} \sum_{j=1}^{m} \mathbf{V}_{jk} \mathbf{P}_a \mathbf{D}_k^{(i)} \mathbf{X}_i \mathbf{u}_{jk} \right) + \frac{\beta}{2} \sum_{k=1}^{P} \sum_{j=1}^{m} \left( \|\mathbf{u}_{jk}\|_F^2 + \|\mathbf{V}_{jk}\|_F^2 \right) \tag{130}$$

*Proof.* The proofs follow almost identically from the proof of 3.7. In the linear case, the convex objective is given by

$$p_{LCNN}^* = \min_{\mathbf{Z} \in \mathbb{R}^{h \times ac}} \sum_{i=1}^{n} F\left( \begin{bmatrix} \text{trace}(\mathbf{P}_a \mathbf{X}_i \mathbf{Z}^{(1)}) \\ \vdots \\ \text{trace}(\mathbf{P}_a \mathbf{X}_i \mathbf{Z}^{(c)}) \end{bmatrix} \right) + \beta \|\mathbf{Z}\|_*. \tag{131}$$

In order to compute the Burer-Monteiro factorization, we factor $\mathbf{Z} = \mathbf{U}\mathbf{V}^\top$, where $\mathbf{U} \in \mathbb{R}^{h \times m}$, $\mathbf{V} \in \mathbb{R}^{ac \times m}$. Then, with $\mathbf{V}^{(c')} \in \mathbb{R}^{a \times m}$. Then,

$$\begin{bmatrix} \text{trace}(\mathbf{P}_a \mathbf{X}_i \mathbf{Z}^{(1)}) \\ \vdots \\ \text{trace}(\mathbf{P}_a \mathbf{X}_i \mathbf{Z}^{(c)}) \end{bmatrix} = \begin{bmatrix} \text{trace}(\mathbf{P}_a \mathbf{X}_i \mathbf{U}\mathbf{V}^{(1)^\top}) \\ \vdots \\ \text{trace}(\mathbf{P}_a \mathbf{X}_i \mathbf{U}\mathbf{V}^{(c)^\top}) \end{bmatrix}$$

$$= \begin{bmatrix} \text{trace}(\mathbf{V}^{(1)^\top} \mathbf{P}_a \mathbf{X}_i \mathbf{U}) \\ \vdots \\ \text{trace}(\mathbf{V}^{(c)^\top} \mathbf{P}_a \mathbf{X}_i \mathbf{U}) \end{bmatrix}$$

$$= \begin{bmatrix} \sum_{j=1}^{m} \mathbf{v}_j^{(1)^\top} \mathbf{P}_a \mathbf{X}_i \mathbf{u}_j \\ \vdots \\ \sum_{j=1}^{m} \mathbf{v}_j^{(c)^\top} \mathbf{P}_a \mathbf{X}_i \mathbf{u}_j \end{bmatrix}$$

$$= \sum_{j=1}^{m} \mathbf{V}_j^\top \mathbf{P}_a \mathbf{X}_i \mathbf{u}_j,$$

where

$$\mathbf{V}_j^\top := \begin{bmatrix} \mathbf{v}_j^{(1)^\top} \\ \cdots \\ \mathbf{v}_j^{(c)^\top} \end{bmatrix} \in \mathbb{R}^{c \times a}. \tag{132}$$

The equivalent Burer-Monteiro formulation thus is given by

$$p_{LCNN}^* = \min_{\substack{\{\mathbf{u}_j \in \mathbb{R}^h\}_{j=1}^m \\ \{\mathbf{V}_j \in \mathbb{R}^{c \times a}\}_{j=1}^m}} \sum_{i=1}^n F\left(\sum_{j=1}^m \mathbf{V}_j \mathbf{P}_a \mathbf{X}_i \mathbf{u}_j\right) + \frac{\beta}{2} \sum_{j=1}^m \left(\|\mathbf{u}_j\|_2^2 + \|\mathbf{V}_j\|_F^2\right). \tag{133}$$

In the gated ReLU case, the convex program is given by

$$p_{GCNN}^* = \min_{\mathbf{Z}_k \in \mathbb{R}^{h \times ac}} \sum_{i=1}^n F\left(\sum_{k=1}^P \begin{bmatrix} \text{trace}(\mathbf{P}_a \mathbf{D}_k^{(i)} \mathbf{X}_i \mathbf{Z}_k^{(1)}) \\ \vdots \\ \text{trace}(\mathbf{P}_a \mathbf{D}_k^{(i)} \mathbf{X}_i \mathbf{Z}_k^{(c)}) \end{bmatrix}\right) + \beta \sum_{k=1}^P \|\mathbf{Z}_k\|_*. \tag{134}$$

In order to compute the Burer-Monteiro factorization, we factor $\mathbf{Z}_k = \mathbf{U}_k \mathbf{V}_k^\top$, where $\mathbf{U}_k \in \mathbb{R}^{h \times m}$, $\mathbf{V}_k \in \mathbb{R}^{ac \times m}$. Then, with $\mathbf{V}_k^{(c')} \in \mathbb{R}^{a \times m}$. Then for each $k$,

$$\begin{bmatrix} \text{trace}(\mathbf{P}_a \mathbf{D}_k^{(i)} \mathbf{X}_i \mathbf{Z}_k^{(1)}) \\ \vdots \\ \text{trace}(\mathbf{P}_a \mathbf{D}_k^{(i)} \mathbf{X}_i \mathbf{Z}_k^{(c)}) \end{bmatrix} = \begin{bmatrix} \text{trace}(\mathbf{P}_a \mathbf{D}_k^{(i)} \mathbf{X}_i \mathbf{U}_k \mathbf{V}_k^{(1)^\top}) \\ \vdots \\ \text{trace}(\mathbf{P}_a \mathbf{D}_k^{(i)} \mathbf{X}_i \mathbf{U}_k \mathbf{V}_k^{(c)^\top}) \end{bmatrix}$$

$$= \begin{bmatrix} \text{trace}(\mathbf{V}_k^{(1)^\top} \mathbf{P}_a \mathbf{D}_k^{(i)} \mathbf{X}_i \mathbf{U}_k) \\ \vdots \\ \text{trace}(\mathbf{V}_k^{(c)^\top} \mathbf{P}_a \mathbf{D}_k^{(i)} \mathbf{X}_i \mathbf{U}_k) \end{bmatrix}$$

$$= \begin{bmatrix} \sum_{j=1}^m \mathbf{v}_{jk}^{(1)^\top} \mathbf{P}_a \mathbf{D}_k^{(i)} \mathbf{X}_i \mathbf{u}_{jk} \\ \vdots \\ \sum_{j=1}^m \mathbf{v}_{jk}^{(c)^\top} \mathbf{P}_a \mathbf{D}_k^{(i)} \mathbf{X}_i \mathbf{u}_{jk} \end{bmatrix}$$

$$= \sum_{j=1}^m \mathbf{V}_{jk}^\top \mathbf{P}_a \mathbf{D}_k^{(i)} \mathbf{X}_i \mathbf{u}_{jk},$$

where

$$\mathbf{V}_{jk}^\top := \begin{bmatrix} \mathbf{v}_{jk}^{(1)^\top} \\ \cdots \\ \mathbf{v}_{jk}^{(c)^\top} \end{bmatrix} \in \mathbb{R}^{c \times a}. \tag{135}$$

The equivalent Burer-Monteiro formulation thus is given by

$$p_{GCNN}^* = \min_{\substack{\{\{\mathbf{u}_{jk} \in \mathbb{R}^h\}_{j=1}^m\}_{k=1}^P \\ \{\{\mathbf{V}_{jk} \in \mathbb{R}^{c \times a}\}_{j=1}^m\}_{k=1}^P}} \sum_{i=1}^n F\left(\sum_{k=1}^P \sum_{j=1}^m \mathbf{V}_{jk} \mathbf{P}_a \mathbf{D}_k^{(i)} \mathbf{X}_i \mathbf{u}_{jk}\right) + \frac{\beta}{2} \sum_{k=1}^P \sum_{j=1}^m \left(\|\mathbf{u}_{jk}\|_F^2 + \|\mathbf{V}_{jk}\|_F^2\right).$$

$$\tag{136}$$

$\square$

### B.3 SELF-ATTENTION

**Lemma B.3.** *The Burer-Monteiro factorization of the convex self-attention problem with gated ReLU and ReLU activations are given as follows.*

$$p^*_{GSA} = \min_{\substack{\mathbf{U}_{jk} \\ \mathbf{V}_{jk}}} \sum_{i=1}^n F\left( \sum_{k=1}^P \sum_{j=1}^m \left( \mathrm{diag}^{-1}(\mathbf{D}_k^{(i)}) \odot (\mathbf{X}_i \mathbf{U}_{jk} \mathbf{X}_i^\top) \right) \mathbf{X}_i \mathbf{V}_{jk} \right) \tag{137}$$

$$+ \frac{\beta}{2} \sum_{k=1}^P \sum_{j=1}^m \|\mathbf{U}_{jk}\|_F^2 + \|\mathbf{V}_{jk}\|_F^2$$

$$p^*_{RSA} = \min_{\substack{\mathbf{U}_{jk} \\ \mathbf{V}_{jk}}} \sum_{i=1}^n F\left( \sum_{k=1}^P \sum_{j=1}^m \left( \mathrm{diag}^{-1}(\mathbf{D}_k^{(i)}) \odot (\mathbf{X}_i \mathbf{U}_{jk} \mathbf{X}_i^\top) \right) \mathbf{X}_i \mathbf{V}_{jk} \right) \tag{138}$$

$$+ \frac{\beta}{2} \sum_{k=1}^P \sum_{j=1}^m \|\mathbf{U}_{jk}\|_F^2 + \|\mathbf{V}_{jk}\|_F^2 \quad \text{s.t. } (2\mathrm{diag}^{-1}(\mathbf{D}_k^{(i)}) - \mathbf{1}\mathbf{1}^\top) \odot (\mathbf{X}_i \mathbf{U}_{jk} \mathbf{X}_i^\top) \geq 0,$$

*where* $\mathrm{diag}^{-1}(\mathbf{D}_k^{(i)}) \in \mathbb{R}^{s \times s}$ *takes elements along the diagonal of* $\mathbf{D}_k^{(i)}$ *and places them in matrix form.*

*Proof.* Courtesy of (Sahiner et al., 2022), we first present the equivalent convex models for gated ReLU and ReLU activation self attention. First, define

$$\mathbf{X} := \begin{bmatrix} \mathbf{X}_1 \otimes \mathbf{X}_1 \\ \cdots \\ \mathbf{X}_n \otimes \mathbf{X}_n \end{bmatrix}$$

$$\{\mathbf{D}_j\}_{j=1}^P := \{\mathrm{diag}\left(\mathbb{1}\{\mathbf{X}\mathbf{u} \geq 0\}\right)\}_{j=1}^P,$$

then, we have

$$p^*_{GSA} = \min_{\mathbf{Z}_j \in \mathbb{R}^{d^2 \times dc}} \sum_{i=1}^n \mathcal{L}\left( \sum_{j=1}^P \sum_{k=1}^d \sum_{\ell=1}^d \mathbf{G}_{i,j}^{(k,\ell)} \mathbf{X}_i \mathbf{Z}_j^{(k,\ell)}, \mathbf{Y}_i \right) + \beta \sum_{j=1}^P \|\mathbf{Z}_j\|_*, \tag{139}$$

$$p^*_{RSA} = \min_{\mathbf{Z}_j \in \mathbb{R}^{d^2 \times dc}} \sum_{i=1}^n \mathcal{L}\left( \sum_{j=1}^P \sum_{k=1}^d \sum_{\ell=1}^d \mathbf{G}_{i,j}^{(k,\ell)} \mathbf{X}_i \mathbf{Z}_j^{(k,\ell)}, \mathbf{Y}_i \right) + \beta \sum_{j=1}^P \|\mathbf{Z}_j\|_{\mathrm{K}_j,*}, \tag{140}$$

where

$$\mathbf{G}_{i,j} := (\mathbf{X}_i \otimes \mathbf{I}_s)^\top \mathbf{D}_j^{(i)} (\mathbf{X}_i \otimes \mathbf{I}_s),$$

for $\mathbf{G}_{i,j}^{(k,\ell)} \in \mathbb{R}^{s \times s}$ and $\mathbf{Z}_j^{(k,\ell)} \in \mathbb{R}^{d \times c}$.

We then proceed to take the Burer-Monteiro factorization of these models. Here, we will show the Burer-Monteiro factorization of the ReLU model, noting that the proof is the same for the Gated ReLU model sans the constraints.

We let $\mathbf{Z}_j = \mathbf{U}_j \mathbf{V}_j^\top$, where $\mathbf{U}_j \in \mathbb{R}^{d^2 \times m}$ and $\mathbf{V}_j \in \mathbb{R}^{dc \times m}$, where $\mathbf{K}_j \mathbf{U}_j \geq 0$. Let $\mathrm{vec}^{-1}(\mathbf{u_{jx}}) \in \mathbb{R}^{d \times d}$ be the result of taking chunks of $d$-length vectors from $\mathbf{u}_{jx}$ for $j \in [m]$ and stacking them in columns. Similarly, let $\mathrm{vec}^{-1}(\mathbf{v_{jx}}) \in \mathbb{R}^{c \times d}$ be the result of taking chunks of $c$-length vectors from $\mathbf{v}_{jx}$ and stacking them in columns. Furthermore, we will let $\mathrm{vec}^{-1}(\mathbf{u_{jx}})_k$ be the $k$th column of $\mathrm{vec}^{-1}(\mathbf{u_{jx}})$. Then, recognize that

$$\mathbf{Z}_j^{(k,\ell)} = \sum_{x=1}^m \mathrm{vec}^{-1}(\mathbf{u_{jx}})_k \mathrm{vec}^{-1}(\mathbf{v_{jx}})_\ell^\top.$$

Thus, for each $j$,

$$\sum_{k=1}^{d}\sum_{\ell=1}^{d}\mathbf{G}_{i,j}^{(k,\ell)}\mathbf{X}_i\mathbf{Z}_j^{(k,\ell)} = \sum_{x=1}^{m}\sum_{k=1}^{d}\sum_{\ell=1}^{d}\mathbf{G}_{i,j}^{(k,\ell)}\mathbf{X}_i\mathrm{vec}^{-1}(\mathbf{u_{jx}})_k\mathrm{vec}^{-1}(\mathbf{v_{jx}})_\ell^\top$$

$$= \sum_{x=1}^{m}\sum_{k=1}^{d}\sum_{\ell=1}^{d}\left[(\mathbf{X}_i\otimes\mathbf{I}_s)^\top\mathbf{D}_j^{(i)}(\mathbf{X}_i\otimes\mathbf{I}_s)\right]^{(k,\ell)}\mathbf{X}_i\mathrm{vec}^{-1}(\mathbf{u_{jx}})_k\mathrm{vec}^{-1}(\mathbf{v_{jx}})_\ell^\top$$

$$= \sum_{x=1}^{m}\sum_{k=1}^{d}\sum_{\ell=1}^{d}(\mathbf{X}_i[\cdot,k]\otimes\mathbf{I}_s)^\top\mathbf{D}_j^{(i)}(\mathbf{X}_i[\cdot,\ell]\otimes\mathbf{I}_s)\mathbf{X}_i\mathrm{vec}^{-1}(\mathbf{u_{jx}})_k\mathrm{vec}^{-1}(\mathbf{v_{jx}})_\ell^\top$$

$$= \sum_{y=1}^{s}\sum_{x=1}^{m}\sum_{k=1}^{d}\sum_{\ell=1}^{d}\mathbf{D}_j^{(i)^{(y,y)}}\mathbf{X}_i\mathrm{vec}^{-1}(\mathbf{u_{jx}})_k\mathrm{vec}^{-1}(\mathbf{v_{jx}})_\ell^\top\mathbf{X}_i[y,k]\mathbf{X}_i[y,\ell]$$

$$= \sum_{x=1}^{m}\left(\mathrm{diag}^{-1}(\mathbf{D}_j^{(i)})\odot(\mathbf{X}_i\mathbf{U}_{jx}\mathbf{X}_i^\top)\right)\mathbf{X}_i\mathbf{V}_{jx},$$

where the constraint that $(2\mathbf{D}_j^{(i)} - \mathbf{I})\mathbf{X}\mathbf{U}_j \geq 0$ can also be re-written as $(2\mathrm{diag}^{-1}(\mathbf{D}_k^{(i)}) - \mathbf{1}\mathbf{1}^\top)\odot(\mathbf{X}_i\mathbf{U}_{jx}\mathbf{X}_i^\top) \geq \mathbf{0}$ for all $x \in [m]$. Thus, we have proven the statement.

$\square$

## C EXPERIMENTAL DETAILS

In all cases, we solve the BM factorization with GD using Pytorch (Paszke et al., 2019) on a CPU with a momentum parameter of 0.9, a learning rate of 1.0 which decays by a factor of 0.9 whenever the training loss plateaus, and train for 20000 epochs such that GD always converges. For convex optimization, to determine the global optimum of each problem, we use the MOSEK interior point solver (Andersen & Andersen, 2000) with CVXPY (Diamond & Boyd, 2016). The parameters we evaluate are $n \in [3, 15, 45, 75, 150]$, $\beta \in [10^{-4}, 10^{-3}, 10^{-2}, 10^{-1}]$ and $m \in [1, 2, 5]$. We use a randomly subsampled set of $\hat{P} = 100$ hyperplane arrangements. We perform this experiment over three random seeds, which are used to generate the hyperplane arrangements as well as the random Gaussian initializations of the weights.

## D ADDITIONAL EXPERIMENTAL RESULTS: BM ENABLES LAYERWISE TRAINING OF CONVEX CNNS

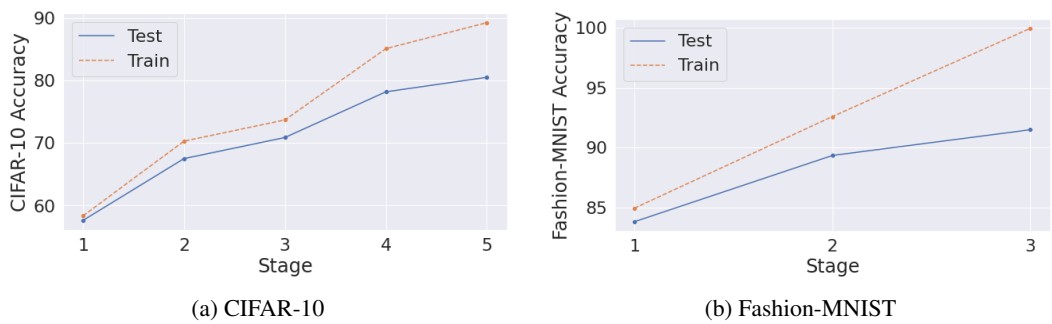

(a) CIFAR-10  (b) Fashion-MNIST

Figure 3: BM enables layerwise training of convex gated ReLU CNNs, which are competitive with end-to-end ReLU networks of similar depth. For CIFAR-10, we achieve a test accuracy of 80.5% compared to 81.6% for end-to-end non-convex training, and for Fashion-MNIST we achieve a test accuracy of 91.5% compared to 91.2% for end-to-end non-convex training (Kiliçarslan & Celik, 2021; Bhatnagar et al., 2017).

We also provide some additional experimental results not presented in the main paper in order to supplement our submission.

We consider the task of leveraging the theory of two-layer convex ReLU neural networks for training deep image classifiers. In particular, following the approach of (Belilovsky et al., 2019), we seek to train two-layer convex CNNs greedily to mimic the performance of a deep network. In the non-convex setting, the greedy approach proceeds by training a single two-layer CNN, then freezing the weights of this CNN, using the latent representation of this CNN as the input features for another two-layer CNN, and repeating this process for a specified number of stages. We leverage the result of Theorem 3.6 to convert this non-convex layerwise training procedure to a convex one, training stages of convex two-layer gated ReLU CNNs with average pooling. We apply this procedure to the image classification datasets of CIFAR-10 (Krizhevsky et al., 2009) and Fashion-MNIST (Xiao et al., 2017), following all general architecture choices from (Belilovsky et al., 2019) (see Appendix for details). We model the scenario in which we are in a memory-limited setting, restricting our memory to a single 12GB GPU.

In this memory-limited setting, layerwise training with the full convex model is impossible, because the nuclear norm penalty requires an SVD at each iteration, and further because the latent representation to be used as input for the second stage, given by $\{\{\mathbf{D}_j^{(i)}\mathbf{X}_i\mathbf{Z}_j^{(c')})\}_{j=1}^P\}_{c'=1}^c$, has $Pac$ channels, which for reasonable choices of $P = 256$, $a = 4$, $c = 10$ yields upwards of $10^4$ channels for the input to the second CNN stage. Accordingly, for this model, we employ the BM factorization with $m = 1$, allowing for a latent representation for the second stage consisting of only $P$ channels.

As we show in Figure 3, this BM scheme for layerwise training allows for both test and train accuracies to improve one stage the next of the layerwise training procedure, reaching the performance of much deeper networks while enabling a convex optimization procedure. In particular, training five stages of a BM factorized convex two-layer gated ReLU CNN on CIFAR-10 resulted in a final test accuracy of 80.5%. Previously, it has been demonstrated that a six-layer ReLU CNN achieves 81.6% on CIFAR-10 when trained end-to-end (Kiliçarslan & Celik, 2021). In addition, the three-stage trained BM factorized convex two-layer gated ReLU CNN on Fasion-MNIST achieved a final test accuracy of 91.5%, compared to 91.2% for a four-layer ReLU CNN trained end-to-end (Bhatnagar et al., 2017). These results demonstrate that the BM factorization is essential for convex neural networks to match the performance of deep end-to-end trained ReLU networks.

Our layerwise training procedure was trained on a single NVIDIA 1080 Ti GPU using the Pytorch deep learning library (Paszke et al., 2019). In particular, we follow the implementation of (Belilovsky et al., 2019), who proposed greedily, sequentially training two-layer CNNs. At each stage, a two-layer CNN (convolutional layer + average pooling + fully connected layer) is trained, and then the weights are frozen, the fully connected layer and average pooling are discarded, and the trained convolutional layer is used as a feature-generator for the following stage. At certain stages, before the CNN is applied, an invertible downsampling operation (Dinh et al., 2016) is used to reduce the spatial dimensions of the image. In (Belilovsky et al., 2019), $m = 128$ convolutional features per stage are used, with ReLU activations as well as an average pooling operation to spatial dimensions of $2 \times 2$ (i.e. $a = 4$) is used, followed by a flattening operation and a fully connected layer. They also use a softmax cross-entropy loss, a batch size of 128, weight decay parameter of $\beta = 5e - 4$, along with stochastic gradient descent (SGD) with momentum fixed to 0.9, 50 epochs per stage, and learning rate decay by a factor of 0.2 every 15 epochs.

In our experiments, we keep all network and optimization parameters the same, aside from replacing the non-convex CNN at each stage with our convex CNN objective (26). We then apply the Burer-Monteiro factorization with $m = 1$ to this architecture to make it tractable for layerwise learning as described in the main paper. At each stage, we randomly subsample $\hat{P} = 256$ hyperplane arrangements, rather than enumerate over all $P$, which is a high-order polynomial in terms of $n$. We further use gated ReLU rather than ReLU activations for simplicity, which can work as well as ReLU in practice (Fiat et al., 2019). These techniques have been used effectively for convex learning to exceed the performance of two-layer non-convex neural networks (Pilanci & Ergen, 2020; Ergen & Pilanci, 2020; Ergen et al., 2021).

For the CIFAR-10 experiment, we use 5 stages (following (Belilovsky et al., 2019)), whereas for the Fashion-MNIST experiment, we use 3 stages, since the training accuracy saturates after 3 stages. We

choose learning rates per stage from $\{10^{-1}, 10^{-2}, 10^{-3}, 10^{-4}\}$ per stage based on training accuracy for CIFAR-10. The chosen learning rates were $[10^{-1}, 10^{-2}, 10^{-3}, 10^{-2}, 10^{-2}]$ for CIFAR-10. For Fashion-MNIST, we empirically observed the training loss was better optimized with slightly higher learning rates, so we used $[2 \times 10^{-1}, 5 \times 10^{-2}, 5 \times 10^{-3}]$. All code used to run our experiments is provided. Ultimately, our CIFAR-10 network with 5 stages took 9163 seconds to train, and the Fashion-MNIST network with 3 stages took 4931 seconds to train. In (Kiliçarslan & Celik, 2021), it is shown that an end-to-end 6-layer CNN with ReLU activations takes 640 seconds to train on CIFAR-10 and 285 seconds to train on Fasion-MNIST. We note that the purpose of this experiment is not to advocate for the use of layerwise BM networks over end-to-end trained networks, but simply to demonstrate the utility of the BM network in enabling convex neural networks to scale to the performance of end-to-end deep networks by using layerwise learning.

