# OpenReview forum: "Scaling Convex Neural Networks with Burer-Monteiro Factorization"
_ICLR.cc/2023/Conference — Submitted to ICLR 2023_

### Official Review · Reviewer_tmAf · 2022-10-24

**Confidence:** 2
**Correctness:** 4
**Technical Novelty And Significance:** 3
**Empirical Novelty And Significance:** 2
**Recommendation:** 6

**Clarity, Quality, Novelty And Reproducibility:**

I realize the ship has already sailed here in the literature, but I find it the term "convex neural network" without reference to _training_ to be confusing (I read it instead as synonymous to an "input convex neural network"). Consider making the reference to the training problem more explicit in the abstract and introduction (as an example, "a wide variety of ...neural networks ... can be posed as equivalent convex optimization problems").

**Strength And Weaknesses:**

The paper does a great job summarizing the prior art in this line of work. The contributions are seemingly novel, and the use of the BM factorization in this context is a smart and natural one.

I think the authors could be a bit clearer in stating the scope of their contributions in the abstract and introduction. It seems that the results of the paper are restricted to _two layer_ MLPs, but the abstract and Section 1.1 do not make this explicit (though some preceding text on page 2 does mention that this).



**Summary Of The Paper:**

This paper presents new algorithms for learning (two-layer) multi-layer perceptrons with ReLU activations. The approach uses the Burer-Monteiro factorization, which yields a nonconvex optimization problem that (under certain low-rank conditions) does not admit spurious local optima. A collection of other related results are applied to other architectures, and brief computational results are provided.

**Summary Of The Review:**

The paper is well-written and self-contained, and is seemingly a natural and worthy contribution to the literature. However, I am not an expert in the subfield studied in the paper, so my degree of confidence in this assessment is relatively low.

---

> ### Author Response · Authors · 2022-11-14
> **Response to Reviewer tmAf**
>
> We would like to thank the reviewer for the feedback and comments. We have added some recommended clarifications mentioned by the reviewer. Please see our responses below.
>
> **Results of the paper are restricted to two-layer MLPs**
>
> To be clear, the results of our paper apply to two-layer MLPs (section 3.1), CNNs (section 3.2), and self-attention networks (section 3.3). We also state in the first sentence of the abstract and Section 1.1 that it has recently been demonstrated that two-layer neural networks have been shown to be equivalent to convex programs, making it clear that two-layer networks are being considered in the scope of this paper.
>
> **Clarity on the term "convex neural networks"**
>
> We thank the reviewer for this note. For clarity, we have updated the abstract and introduction to explicitly state that the training problem of two-layer neural networks are posed as equivalent convex optimization problems.

---

### Official Review · Reviewer_NEnW · 2022-10-25

**Confidence:** 2
**Correctness:** 4
**Technical Novelty And Significance:** 3
**Empirical Novelty And Significance:** 2
**Recommendation:** 5

**Clarity, Quality, Novelty And Reproducibility:**

Clarity:  This looks like a journal paper that was rammed into a conference paper.  There are way too many details that require verification, and it isn't clear why all of the different types of networks needed to be presented.  One well-worked example would have been sufficient for a conference submission (at least in my opinion), e.g., the MLP seems to have sufficient novelty and it is the one used in the experiment.  Or do the technical details vary dramatically between the different types of networks?  Other than breadth, I'm not sure that I see what is gained by including them all.

Quality:  The experimental section is really a bit weak.  Too many theoretical results left little space in the main text for a more detailed experimental evaluation.  In particular, the experiment in the main text is only on synthetic data.

Novelty:  To the best of my knowledge, the application of the Burer-Monteiro factorization approach appears to be novel in this setting.

Reproducibility:  While the experimental details have mostly been relegated to the supplementary material, it does seem that there are sufficient details to reproduce the results.

**Strength And Weaknesses:**

Strengths:  The general approach appears to be quite applicable to a wide variety of NN architectures.

Weaknesses:  There are too many NNs considered at the expense of more detailed experimental evaluation.  The main text just reads like a laundry list of similar results that don't seem to add any depth to the discussion.  In its current form, this work feels more appropriate for journal publication.

**Summary Of The Paper:**

The authors describe an application of Burer-Monteiro factorization to convex programs on neural networks.  The aim is to produce a non-convex, but tractable, optimization problem that, under certain conditions, do not have spurious local minima.  The approach is applied on a variety of NN architectures and validated with a simple synthetic experiment.

**Summary Of The Review:**

The paper seems to provide novel approaches and insights, but it is plagued by too much information in too short a space.

---

> ### Author Response · Authors · 2022-11-14
> **Response to Reviewer NEnW**
>
> We would like to thank the reviewer for the feedback and comments. We hope that you would consider increasing your score if your concerns are adequately addressed.
>
> **Too many NNs considered**
>
> A core insight of this work is that a common technique (the Burer-Monteiro factorization) can be applied to a wide variety of convex neural network architectures. Burer-Monteiro factorizations have traditionally only been applied to the MLP setting of neural networks, and in very restricted settings. Thus, a major contribution of this work is to extend this analysis to more commonly used neural network architectures, such as CNNs and multi-head self-attention networks. For example, we are the first to show that linear multi-head self-attention networks have no spurious local minima in the case that they have sufficiently many heads. These are important contributions that we believe are important to highlight.
>
> **Too little experimental results**
>
> Aside from the experimental result in the main paper, we also provide a larger scale experimental result on the BM factorization in Appendix D. These results demonstrate how the BM factorization enables layer-wise learning of convex CNN which is otherwise intractable. We show that this layerwise training procedure matches the performance of end-to-end trained CNNs of the same size, and thus that the BM factorization provides further practical utility.

---

### Official Review · Reviewer_jgFt · 2022-11-02

**Confidence:** 3
**Correctness:** 3
**Technical Novelty And Significance:** 3
**Empirical Novelty And Significance:** 3
**Recommendation:** 8

**Clarity, Quality, Novelty And Reproducibility:**

The paper contains a number of interesting elements but it has to be rewritten. See my detailed comments below

**Strength And Weaknesses:**

The paper is interesting. My main concern is with the readability/organization of the paper. I feel that it would improve readability a lot if you were just removing some of the discussion between your introduction of the formulations p_{LMLP}, p_{GMLP} and p_{RMLP} and their corresponding Burer Monteiro factorizations. I would suggest compressing (if not completely removing) section 1.2 and expand on Lemma 3.1 and Theorem 3.4. providing diagrams (As an example in Corollary 3.8.1. it is hard to picture the network.) for some of the models discussed to clarify the meaning of the variables appearing in each of the optimisation formulations (e.g. 28/29).

Also after reading the whole paper, instead of introducing the convex programs in section 2.1. and then discuss their BM factorisations in separate later subsections, I would replace section 2.1. with a short general introduction to the whole paper and then introduced the convex formulation along each of their BM factorisations. This will make the reading easier by eliminating the need to go back and forth between 2.1 and pages 6/7/8 .

From the space you will spare by eliminating section 1.2, you will also gain space to discuss and improve the numerical experiments.

**Summary Of The Paper:**

The paper discusses the use of the Burer Monteiro factorization in the setting of convex neural network. The main result of the paper are a number of theoretical guarantees on the convergence of gradient descent (i.e. absence of saddles) for appropriate choices of a variety of parameters such as the size of the hidden layer, number of training examples and rank of the Burer Monteiro factorization. The authors tackle a number of architectures from the simple multilayer fully connected perceptron defined on a variety of activation functions (linear , Relu, gated Relu), convolutional neural networks and self attention networks and derive corresponding convergence guarantees for each architecture.

**Summary Of The Review:**


page 3

- You never introduce F. I assume this corresponds to the loss? l2 loss ?

- Aren’t the D_j matrices redundant? given that the factorization enforced by the quasi nuclear norm requires Ku to be non negative? I.e. this also seems to be suggested by your expression of D_j X Z in the paragraph below equation (4)

- What does e stand for in the bound on P? Is that ln(1) ?

page 4

- When you introduce the convex formulation for the Relu based convolutional neural networks, it is not clear what X_i and in particular h and K represent. Is the matrix X_i reorganizing the prototype x_i as a circulant matrix?

- Again, in the paragraph above Equation (8), you use the notation n. I guess ’n’ refers to the input dimension as before. Is c the hidden layer size? you should say it more clearly
- The condition m>=m^* on the one hand and m^*<= nc on the other seem a little too simple to me. I.e. take m^* = 1 and you get unconditional convexity?
- When you write “Since P is exponential in the rank of X, for fixed filter size h, P is polynomial in all other problem dimension” do you mean “Although P is exponential…” ?  If the D_j’s are the same matrices as those you introduce in (3), why not getting rid of this line and adding a reference to the bound on P which you provide below (3), indicating that in the case of  a convolutional network the dependence on n reduces to a dependence on h?
- In the paragraph above equation (10). Again, I have some concern with your bounds on m as they seem to suggest that the equivalence holds for any value of m

page 5

- The sentence “However, if the gradient of f is not Lipschitz-continuous, there are no guarantees that gradient descent will find a second-order critical point of (12): one may encounter a stationary point which is a saddle” which comes at the end of the parameter above Equation (13) should come earlier. When you cite the result of Bach according to which all local mins are global mins, you should indicate that convergence to those minimas is not necessarily guaranteed.

page 6

- At the beginning of the page, what do \mathcal{B}_C and \mathcal{B}_R stand for here? I guess this is related to the corresponding norms in (16)
- I don’t see the link between the variational formulation you introduce in (17) and the regularization terms appearing in each of your formulations (19), (20), (21). In particular in (21), what do the sets \mathcal{B}_C and \mathcal{B}_R represent?
- It is not clear to me how formulation (3) can be written equivalently as (21). In particular, if the original formulation (3) is NP hard to solve, how can the Burer Monteiro factorization become tractable? Either you have poorer/lose convergence guarantees, or you must have some conservation of complexity. Some more details would be welcome
- At the end of the page, in the last paragraph, why would it make sense to take X = I? Isn’t X used to represent the feature matrix ?
- Equation (22), Either you add the hat on the Z subscript of the gradient, or you remove it from the argument of F

page 7
- The transition from (22) to (23) in the case of the Relu is not clear. What does \mathcal{B}_2 mean in this case ?
- if you look at (21) the K_j matrix multiplies U_j (the left factor of Z) while in (23) it multiplies a vector u, which from the constraint u \in \mathcal{B}_2  seems to be the vector appearing in the definition of the D norm in (17).. This is not clear
- I would remove the paragraph “We should note that some stationary points are clearly present in any problem, such as (Uˆ , Vˆ ) = (0, 0), so we cannot conclude that all stationary points are global optima. However, in certain cases, the optimality gap of stationary points (22) is always zero as we show next. ” which is unclear

page 8
- In the statement of Corollary 3.7.1. What is the meaning of Z^{(j)}_{k’} ? What is the meaning of \mathcal{B}_2 ?
- In the statement of Lemma 3.8
- In the statement of Corollary 3.8.1, it would be useful to recall the meaning of d,c,m. d is the number of features. How about c, m ?
- In the statement of Corollary 3.8.1. Again, I wondering if what you mean is not m^*\geq min\left\{d^2, cd\right\}?

page 9

- The plots in figure 2 look very similar. Also from formulation (20) which seems to be the one you use, using d=2, c=3, does not look like a very challenging scenario. In fact you know you are in the regime in which all local minimas are also global. Although I know there can be saddles as well, it might be more interesting to study the evolution of the optimality gap for larger values of n.
- “We find that GD applied to the BM factorization finds “subtle" saddle points: not quite lo- cal minima, but close”. Do you mean that optimality gaps are small but not small enough to ensure a local minimum ?
- “there is only a minor relationship between the optimality gap and the size of the BM factorization m ”. I would replace “size” by rank (which seems more appropriate to me here). Also when you say "there is only a minor relationship between the optimality gap and the size of the BM factorization m”, it seems to me that the optimality gap is independent from pretty much all the quantities you consider except for the regularization. You should be able to get the analytic expression of the gap. How does this expression vary as a function of m, n? This should be further discussed.
- The last paragraph of section 4 is not very clear. In particular, I don’t see why the experiments particularly highlight the need to consider saddles (i.e. in fact if you look at the plots, you see that for m>max(c,d) the gradient iterates converge to the global minimum of the loss. To me, the relevance of saddles would be illustrated if the bound was non zero in that regime where all local mins are also guaranteed to be global mins).

Additional typos:
- page 6, first line : “provides a novel result in the form of a optimality gap bound” —> “aN optimality gap bound”
- same page, next sentence, “To our knowledge, this is the first result of that generalizes the optimality” —> “the first result that generalizes”
- page 7, below equation (24) “we can be assured” —> “we can ensure”

---

> ### Author Response · Authors · 2022-11-14
> **Response to Reviewer jgFt**
>
> We would like to thank the reviewer for the detailed feedback and comments. We have updated many typos mentioned by the reviewer, and some suggested cuts for space. Please see the detailed clarifications to certain points below.
>
> **page 3**
>
> $F$ is defined in Section 2 under Preliminaries on page 3, where we state, "Unless otherwise stated, let F be a convex, differentiable function."
>
> It isn't clear what is meant by D_j matrices being redundant. These D_j matrices inform the K_j matrix which creates the halfspace in which the quasi nuclear norm operates.
>
> Yes, e is Euler's number.
>
> **page 4**
>
> In the CNN formulation, X_i is a re-arranged data matrix where each patch that the convolutional kernel is exposed to is flattened into an individual row of dimension $h$, which is given by the product of the kernel matrix dimensions. $K$ is the number of such patches, which depends on kernel size, padding size, stride, and dilation. We have clarified this in the revision.
>
> Yes, $n$ is the number of samples and $c$ is the output dimension (e.g. number of classes for multi-class classification). We have clarified this in the revision in Preliminaries.
>
> For certain problems, it may be that $m^* = 1$. $m^*$ is an unknown variable that denotes the sparsity of the optimal solution, which is dependent on $F$, $X$, and $\beta$. For example, as $\beta \to \infty$, the optimal solution is the zero network, and thus $m^* = 0$. We have clarified the problem-dependence of $m^*$ in the revision.
>
> The point of $P$ is well taken. We have updated accordingly.
>
> **page 6**
>
> $\mathcal{B}_C$ and $\mathcal{B}_R$ are defined in Preliminaries (Section 2). In (21), $\mathcal{B}_C := \{u: \|u\|_2 \leq 1, (2D_j - I_n) Xu_j \geq 0\}$ for each $j$.
>
> Both (3) and (21) are NP-hard to solve. While the Burer-Monteiro factorization is tractable and it is more easy to analyze (say for optimality conditions of stationary points), there is no guarantee of finding the global optimum.
>
> $X$ = $I$ is the case where one wants to perform matrix factorization, rather than a standard neural network.
>
> In equation 22, we take the gradient w.r.t variable $Z$ and evaluate at $\hat{Z}$.
>
> **page 7**
>
> $\mathcal{B}_2$ is the unit $\ell_2$ ball, as we defined in the Preliminaries. We simply in (23) are substituting the definition of linear operator $M$ and convex regularizer $D$ in the case of the ReLU neural network.
>
> $u \in \mathcal{B}_C$ appears in (17). In the case of a ReLU neural network, as explained above,  $\mathcal{B}_C := \{u: \|u\|_2 \leq 1, K_ju_j \geq 0\}$ for each $j$.
>
> **page 8**
>
> $Z_k^{(j)}$ is the $j$th block of appropriate size of matrix $Z_j$, as we define in the Preliminaries (Section 2).
>
> As we state in the Corollary statement, $m$ is the number of self-attention heads. We have clarified in Section 2 that $c$ is the number of output classes.
>
> **page 9**
>
> For $m =5$, in every case, the optimality gap is non-zero except when $\beta = 0.1$. This indicates that the gradient iterates when m > max(c, d) do not always converge to the global minimum. This is why we conclude that there are saddle points, since in this regime all local minima are guaranteed to be global minima.

---

### Official Review · Reviewer_JjJ3 · 2022-11-04

**Confidence:** 4
**Correctness:** 3
**Technical Novelty And Significance:** 2
**Empirical Novelty And Significance:** 1
**Recommendation:** 6

**Clarity, Quality, Novelty And Reproducibility:**

The paper is clear, but the results are very incremental and with null practical impact.

**Strength And Weaknesses:**

I see several weaknesses:

- The improvement is marginal. There are such results of tens of types of 2 layer neural networks (see the 9 cited, all in the last two years by Pilanci, Ergen, Sahiner and coauthors). This is yet another paper exploiting the same idea, thus the incrementality of the proposed approach.
- The focus is on 2 layer neural networks, which contradicts the claims about "useful practical information". Can the authors point relevant practical papers in the literature where linear 2 layer networks are used?
- The experiments are in dimension 2, which tends to indicate that the methods scales very badly with the dimension/rank of the data, as all convex reformulation methods for neural networks. It seems to me the benefits of the method, compared to SGD, are only on toy setting like this one, and the method becomes impractical in real life scenarios.

**Summary Of The Paper:**

It has been shown by Ergen and Pilanci (2020) that 2 layer relu neural networks with squared L2 regularization are equivalent to regularized convex problems. The latter convex problems are in extremely high dimension and thus not tractable.
The regularization in the equivalent convex problems encourages low rank. hence, the paper proposes to use a Burer-Monteiro factorization to solve them.


**Summary Of The Review:**

Incremental contribution to the field of convex reformulations of neural networks. Experiments are in dimension 2, and models limited to 2 layers.

I would update my score if the rebuttal included an experiment with the number of samples and the number of features in the thousands (without the data matrix being low rank).

---

> ### Author Response · Authors · 2022-11-14
> **Response to Reviewer JjJ3**
>
> We would like to thank the reviewer for the feedback and comments. We hope that you would consider increasing your score if your concerns are adequately addressed.
>
> **Marginal Improvements over Previous Works**
>
> While there are many works on convex two-layer neural networks, as we state in Section 1.2, no existing approaches for solving them are effective for training at large scales. The BM approach allows for more scalable solutions--by reducing the per-iteration complexity to that of a standard non-convex neural network. Our experiments in Appendix D on layerwise training further illustrate that our proposed approach is much more scalable than previous convex neural network approaches.
>
> **Uses of two-layer networks**
>
> To be clear, the results of our paper do not only apply to linear two-layer networks, but also those with Gated ReLU and ReLU activation. Two-layer networks are actually still widely used in many settings, and have practical importance. For example, the widely used word embedding model word2vec is composed of a two-layer MLP [1]. Many transfer learning benchmarks use two-layer MLPs on top of frozen, pre-trained backbones, a practice referred to as ``probing" and proposed originally in [2] and [3]. This approach has been used extensively in literature, such as for the methods in [4], [5], [6] and [7], as well as PAPA [8] and CLIP-Adapter [9]. Lastly, many graphics use cases use shallow MLPs, such as Instant NGP [10] and TensoRF [11].
>
> **Scalability of Experiments**
>
> The experiments in Section 4 are intended to be precise and interpretable, which is why a low-dimensional dataset was chosen. The per-iteration complexity of the BM factorization is equivalent to that of standard non-convex neural networks. The intention of the work is to provide a scalable and interpretable method for training convex neural networks, which the BM approach provides. Our experiments in Appendix D on layerwise training take on the realistic multi-layer CNN task with much higher dimension, which is not tractable for convex neural networks as proposed in previous works. This is applied to the CIFAR-10 and Fashion-MNIST datasets, which are composed of 50k and 60k samples of dimension 3072 and 784 respectively.
>
>
> **References**
>
> [1] Distributed Representations of Words and Phrases, and their Compositionality (Mikolov et al., 2013)
>
> [2] Fine-grained analysis of sentence embeddings using auxiliary prediction tasks (Adi et al., 2016)
>
> [3] What do you learn from context? Probing for sentence structure in contextualized word representations (Tenney et al., 2019)
>
> [4] Probing Linguistic Information For Logical Inference In Pre-trained Language Models (Chen et al., 2022)
>
> [5] On the data requirements of probing (Zhu et al., 2022)
>
> [6] Does Vision-and-Language Pretraining Improve Lexical Grounding? (Yun et al., 2021)
>
> [7] Quantifying the contextualization of word representations with semantic class probing (Zhao et al., 2020)
>
> [8] How Much Does Attention Actually Attend? Questioning the Importance of Attention in Pretrained Transformers (Hassid et al., 2022)
>
> [9] CLIP-Adapter: Better Vision-Language Models with Feature Adapters (Gao et al., 2021)
>
> [10] Instant Neural Graphics Primitives with a Multiresolution Hash Encoding (Müller et al., 2022)
>
> [11] TensoRF: Tensorial Radiance Fields (Chen et al., 2022)

---

### Official Review · Reviewer_f4LA · 2022-11-04

**Confidence:** 2
**Correctness:** 3
**Technical Novelty And Significance:** 3
**Empirical Novelty And Significance:** 2
**Recommendation:** 6

**Clarity, Quality, Novelty And Reproducibility:**

The clarity is good, the problems and results are clearly presented, and relate works and techniques are also discussed well enough. The quality and originality are OK, since most results are from using BM factorization on existing equivalent convex optimization problems of NNs, and some results are not known before.

**Strength And Weaknesses:**

**Strength**:

1. The idea of using BM factorization to improve the computational efficiency of equivalent convex problems of two-layered NNs is interesting, and the results seem novel.
2. The presentation of results is clear and easy to follow.
3. Experimental results can verify the proposed BM factorizations.

**Weaknesses**:

1. The neural networks studied in this work are not practical, mainly two-layered MLPs and CNNs. This weakness is from existing equivalent convex optimization problems are for two-layered models.
2. The results for self-attention networks are interesting. However, it has a similar problem of using linear activation, which makes it questionable how useful those results are in practice.
3. The characterizations of relative gaps and conditions for stationary points being global minimizers are presented without explanation, making the implications not clear to me, e.g., the gradient norm in Eq. (23), and the trace conditions in Eq. (28). Could the authors provide some intuitions or detailed explanations for the audience to better understand the meaning of those results?
4. The comments of BM factorization finds "subtle" saddle points at the end of Section 4 is interesting. However, it is not clear to me how is it concluded that from the results in Figure 2 that the found solutions are saddle points rather than local minima, and is there a calculation to verify that?

**Summary Of The Paper:**

This work studies the problem of formulating several (non)linear two-layer neural networks (NNs) as equivalent convex optimization problems, where existing results of equivalent convex optimization problems are computationally expensive.

This work proposes to use Burer-Monteiro (BM) factorization to obtain equivalent and computationally tractable non-convex alternative (rather than convex optimization problems) with no spurious local minima.

Section 3.1 first calculates BM factorization for convex MLP with linear, gated ReLU, and ReLU activations, as shown in Eqs. (19) - (21). Theorem 3.4 characterize the relative optimality gap in and show that under some rank conditions the stationary points of BM factorization  in Eq. (16) are global minimizers of the equivalent convex optimization problem of Eq. (18), meaning that solving stationary points Eqs. (19) - (21) for those NNs could obtain global minimizers of their existing equivalent convex optimization problems.

Section 3.2 first characterizes two-layer convolutional neural networks (CNNs) with arbitrary linear pooling operations as a convex program (Theorem 3.6). Lemma 3.7 gives BM factorization of the convex CNN problem with ReLU activation. Corollary 3.7.1 gives a result for stationary points of Eq. (27) are global minimizers of Eq. (26), under the condition of Eq. (28).

Section 3.3 then studies self-attention networks. Lemma 3.8 shows the BM factorization of the the convex self-attention problem with linear activation (gated ReLU and ReLU activations are also calculated in the appendix), and Corollary 3.8.1 shows that Eq (29) has no spurious local minima as long as the number of heads is large enough.

Section 4 uses experimental results on synthetic dataset, with a gated-ReLU two-layer MLP; as well as (shown in the appendix) on CIFAR-10 and Fashion-MNIST datasets, with  two-layer gated ReLU CNNs. The results show that BM factorization enables layer-wise training of two-layered MLPs and CNNs, achieving comparable results to end-to-end training of networks.

**Summary Of The Review:**

The idea of using BM factorization to improve the computational efficiency of equivalent convex problems of two-layered NNs is reasonable and novel to me. However, most of the results are for two-layered NNs and restricted models like linear activations in self-attention networks, which makes it questionable how useful those results could be in practical NNs.



=====UPDATE=====

I would like to thank the authors for the feedback. I increased my score since it addressed my questions well.

---

> ### Author Response · Authors · 2022-11-14
> **Response to Reviewer f4LA**
>
> We would like to thank the reviewer for the feedback and comments. We hope that you would consider increasing your score if your concerns are adequately addressed.
>
> **Two-layer MLPs and CNNs are not practical**
>
> Two-layer MLPs and CNNs are actually still widely used in many settings, and have practical importance. For example, the widely used word embedding model word2vec is composed of a two-layer MLP [1]. Many transfer learning benchmarks use two-layer MLPs on top of frozen, pre-trained backbones, a practice referred to as ``probing" and proposed originally in [2] and [3]. This approach has been used extensively in literature, such as for the methods in [4], [5], [6] and [7], as well as PAPA [8] and CLIP-Adapter [9]. Lastly, many graphics use cases use shallow MLPs, such as Instant NGP [10] and TensoRF [11].
>
> **Self-attention results only apply to linear activation**
>
> We note in footnote 2 on page 8 that we examine gated-ReLU and ReLU activation multi-head self-attention networks in the Appendix. Please refer to Lemma B.3. in the appendix for equivalent Burer-Monteiro factorizations for self-attention networks. Accordingly, the results of Theorems 3.3, 3.4, and 3.5 apply to gated-ReLU and ReLU activation self-attention networks as well.
>
> **Characterizations of optimality gaps for stationary points not explained**
>
> We now include additional explanation of what the bound in (23) represents on page 6. As for the trace condition in (28), it is simply the substitution of the definition of the convex CNN into $F$.
>
> **Subtle saddle points v.s. local minima**
>
> We direct the reviewer to our discussion on page 9, where we explain for this experimental setup, as long as $m \geq 2$, all local minima are global minima, which is well established in the literature. However, we see that in Figure 2, even when $m=2$ and $m=5$, GD applied to the BM objective does not find the global minimum. It can then be concluded that since the GD soluiton is at a stationary point which is not a global (and therefore not a local) minimum, the GD solution must be at a saddle point.
>
>
> **References**
>
> [1] Distributed Representations of Words and Phrases, and their Compositionality (Mikolov et al., 2013)
>
> [2] Fine-grained analysis of sentence embeddings using auxiliary prediction tasks (Adi et al., 2016)
>
> [3] What do you learn from context? Probing for sentence structure in contextualized word representations (Tenney et al., 2019)
>
> [4] Probing Linguistic Information For Logical Inference In Pre-trained Language Models (Chen et al., 2022)
>
> [5] On the data requirements of probing (Zhu et al., 2022)
>
> [6] Does Vision-and-Language Pretraining Improve Lexical Grounding? (Yun et al., 2021)
>
> [7] Quantifying the contextualization of word representations with semantic class probing (Zhao et al., 2020)
>
> [8] How Much Does Attention Actually Attend? Questioning the Importance of Attention in Pretrained Transformers (Hassid et al., 2022)
>
> [9] CLIP-Adapter: Better Vision-Language Models with Feature Adapters (Gao et al., 2021)
>
> [10] Instant Neural Graphics Primitives with a Multiresolution Hash Encoding (Müller et al., 2022)
>
> [11] TensoRF: Tensorial Radiance Fields (Chen et al., 2022)

---

### Decision · Program_Chairs · 2023-01-20

**Decision:**

Reject

**Justification For Why Not Higher Score:**

For the reasons mentioned in the meta-review

**Justification For Why Not Lower Score:**

N/A

**Metareview: Summary, Strengths And Weaknesses:**

- Summary:

This work studies the problem of formulating several (non)linear two-layer neural networks (NNs) as equivalent convex optimization problems, where existing results of equivalent convex optimization problems are computationally expensive. The paper proposes to use Burer-Monteiro (BM) factorization to obtain equivalent and computationally tractable non-convex alternative (rather than convex optimization problems) with no spurious local minima.

- Strengths:

1. BM factorization in convex NN training is an interesting combination
2. Applicability to a wide range of (shallow) NN architectures

- Weaknesses:

1. Many of the results seem incremental to existing work (several reviewers made such a comment). The AC read the paper + read the discussions and this concern remains.
2. The title of the paper mentions "Scaling"; however, the experimental results are extremely low dimensional to reveal that the methodology really scales and how useful it is in large scale scenarios. This comment has been raised by several reviewers. Even if the goal is to illustrate that the methodology just works + some theoretical results, the aim of the paper is to use BM for scalability, and this is not shown in the current version of the paper.
3. The paper has a lot of results (either already published or contributions of this paper) that mostly blurs the overall flow of the paper, rather than help understanding better the concept and further intuitions might there be. Currently, parts of the paper read as a "collage" of existing results.

- Recommendation:

First of all, the authors should be stayed assured that there were discussions regarding the paper, and the area chair read carefully i) the paper, ii) the reviews, and iii) the discussions after the rebuttal. We acknowledge that the revised paper addresses concerns raised, but the reviewers have remaining concerns.

We recommend the authors include a more thorough experimental validation for their methodology proposal into higher dimensions.  Also, more intuitions and a more clear story (where previous existing works can be deferred in the appendix to save space for more discussion) would be more appreciated. As an area chair, I found the concerns raised by reviewers important to be handled, but I would also consider this case a fortunate one, since this is definitely a constructive feedback to your paper.

We recommend the authors to consider these (common over all reviewers) concerns to revise their paper; we strongly support for a resubmission to a near-future ML conference.